# Small-scale topography explains patterns and dynamics of dissolved organic carbon exports from the riparian zone of a temperate, forested catchment

Benedikt J. Werner[1], Oliver J. Lechtenfeld[2], Andreas Musolff[1], Gerrit H. de Rooij[3], Jie Yang[1], Ralf Gründling[3], Ulrike Werban[4], Jan H. Fleckenstein[1, 5]

[1]Department of Hydrogeology, Helmholtz Centre for Environmental Research – UFZ, 04318 Leipzig, Germany
[2]Department of Analytical Chemistry, Research group BioGeoOmics, Helmholtz Centre for Environmental Research – UFZ, 04318 Leipzig, Germany
[3]Department of Soil System Sciences, Helmholtz Centre for Environmental Research – UFZ, 06120 Halle, Germany
[4]Department of Monitoring and Exploration Technologies, Helmholtz Centre for Environmental Research – UFZ, 04318 Leipzig, Germany
[5] Hydrologic Modelling Unit, Bayreuth Center of Ecology and Environmental Research (BayCEER), University of Bayreuth, Bayreuth, Germany

*Correspondence to*: Benedikt J. Werner (benedikt.werner@ufz.de)

**Abstract.** Export of dissolved organic carbon (DOC) from riparian zones (RZs) is an important component of temperate catchment carbon budgets, but export mechanisms are still poorly understood. Here we show that DOC export is predominantly controlled by the micro-topography of the RZ (lateral variability), and by riparian groundwater level dynamics (temporal variability). From February 2017 until July 2019 we studied topography, DOC quality as well as water fluxes and pathways in the RZ of a small forested catchment and the receiving stream in central Germany. The chemical classification of the riparian groundwater and surface water samples (n = 66) by Fourier-transform ion cyclotron resonance mass spectrometry revealed a cluster of plant-derived, aromatic, and oxygen-rich DOC with high concentrations ($DOC_I$) and a cluster of microbially processed, saturated, and hetero-atom enriched DOC with lower concentrations ($DOC_{II}$). The two DOC clusters were connected to locations with distinctly different values of the high-resolution topographic wetness index ($TWI_{HR}$; @ 1 m resolution) within the study area. Numerical water flow modelling using the integrated surface subsurface model HydroGeoSphere revealed that surface runoff from high $TWI_{HR}$ zones associated with the $DOC_I$ cluster ($DOC_I$ source zones) dominated overall discharge generation and therefore DOC export. Although corresponding to only 15 % of the area in the studied RZ, the $DOC_I$ source zones contributed 1.5 times the DOC export of the remaining 85 % of the area associated with $DOC_{II}$ source zones. Accordingly, DOC quality in stream water sampled under five event flow conditions (n = 73) was closely reflecting the $DOC_I$ quality. Our results suggest that DOC export by surface runoff along dynamically evolving surface flow networks can play a dominant role for DOC exports from RZs with overall low topographic relief and should consequently be considered in catchment scale DOC export models. We propose that proxies of spatial heterogeneity such as the $TWI_{HR}$ can help to delineate the most active source zones and provide a mechanistic basis for improved model conceptualization of DOC exports.

# 1 Introduction

Dissolved organic carbon (DOC) in streams and rivers is in itself of central ecological importance (Cole et al., 2007; Battin et al., 2008), but the amount and quality of DOC also shape water quality through interactions and co-export with other chemicals in terrestrial solute source areas (Ledesma et al., 2016; Sherene, 2010), rivers and lakes (Prairie, 2008). Beside the ecological impacts this alteration may also affect safety and costs of drinking water production (e.g. Wang et al., 2017). Changes in land use, climate and biogeochemical boundary conditions have increased DOC concentrations in surface waters

and altered the quality of the exported DOC in the last decades (Larsen et al., 2011; Chantigny, 2003; Wilson and Xenopoulos, 2008). Routine management of DOC could therefore help to comply with water quality directives and lower the cost of drinking water purification (Matilainen et al., 2011), but understanding how DOC changes and moves within and across ecosystem interfaces, thus linking aquatic and terrestrial carbon cycles are still large knowledge gaps (Butman et al., 2018; Drake et al., 2018; Vachon et al., 2021) that so far impede proper DOC management (Stanley et al., 2012).

Lower order streams make up a large fraction of the total river networks worldwide (Raymond et al., 2013) and their riparian zones (RZs) represent a main source for terrestrial DOC export (Ledesma et al., 2015; Musolff et al., 2018). Therefore RZs of lower order streams – as terrestrial-aquatic interfaces – constitute a general control unit, qualifying them as potential targets for DOC export management. However, describing and quantifying the effective DOC export from such RZs at larger management-scale still remains a challenge due to the high spatio-temporal variability of local mechanisms that control DOC

accumulation, mobilization and transport in RZs (Pinay et al., 2015; Bernhardt et al., 2017; Krause et al., 2014). Large uphill contributing areas deliver a continuous supply of water to the RZ, leading to generally moist conditions with high groundwater levels, even during dry periods. Here, DOC accumulation rates are highest during anaerobic conditions at low temperatures due to low mineralization rates, whereas high mineralization rates can be realized in warmer, oxygenated zones in the soil (Luke et al., 2007). On the other hand, the amount of accumulated DOC and ultimately its export is also dependent on

hydrological connection of DOC sources to the stream. Micro-topography in RZs can induce hot spots of biogeochemical activity (Frei et al., 2012) that contribute disproportionally to nutrient turnover. Depressions in micro-topography collect surficial water (Frei et al. 2010, Scheliga et al., 2019). If these puddles grow to connect with each other, continuous but possibly short-lived surface flow channels can develop that can connect hot spots of DOC production in the shallow soil layers of the RZ to the stream and carry DOC to the stream (during so called hot moments). Therefore, micro-topography in the RZ

is considered a fundamental organizing structure, not only for soil chemistry (Diamond et al., 2020) but also of hydrological connectivity (Frei et al. 2010, Scheliga et al., 2019) that can induce high spatio-temporal heterogeneity of DOC exports. Riparian topography and the dynamics of groundwater levels in the RZ thus can be key drivers of the spatio-temporal patterns of DOC export from RZs.

Several attempts have been made to characterize and quantify the dynamics of runoff generation in RZs and the associated

variability of DOC transport to streams. However, to date model conceptualizations have mainly focused on the vertical

distribution of DOC sources in the subsurface and to a lesser degree on horizontal heterogeneity induced by topography. For instance the dominant source layer concept (Ledesma et al., 2015) focuses on depth-dependent differences in DOC pools in distinct soil layers of a boreal catchment. The dominant source layer concept is based on the transmissivity feedback mechanism (Bishop et al., 2004), which accounts for depth-dependent differences in hydraulic conductivities of soils and the resulting changes in the transmissivity of the soil profile under changing groundwater levels. This concept is taken up in the riparian profile flow-concentration integration model (Rim, Seibert et al., 2009) to model stream solute variability as a function of a non-linear vertical distribution of pore water solute concentrations in riparian soils. Frei et al. (2010, 2012) were able to simulate the complex effects of riparian micro-topography on runoff generation and the formation of biogeochemical hotspots in the subsurface, but their explorative model was computationally expensive and did not explicitly consider DOC transport.

To date variations in the lateral hydrological connectivity of a RZ to a stream have mainly been conceptualized in the context of spatially lumped catchment DOC export models by defining different source zones with variable activation (Dick et al., 2015), largely ignoring small-scale spatio-temporal variability in DOC export from individual, small landscape units (Ledesma et al., 2018a; Dick et al., 2015). Our study aims at bridging the gap between small-scale mechanistic understanding of DOC mobilization and export and its larger-scale, lumped description in most conceptual models.

Improved understanding of the dominant mechanisms of DOC generation and transport in small-scale landscape elements could help to find accessible proxies that can better describe the larger-scale effective DOC-export behavior of catchments (Grabs et al., 2012). Currently existing proxies are mainly based on landscape-scale characteristics like different land use types (Pisani et al., 2020), hydromapping based on convergence of topography (Laudon et al., 2016; Ploum et al., 2020) or general topographic wetness (Musolff et al., 2018; Fellman et al., 2017; Andersson and Nyberg, 2009) in boreal and temperate catchments e.g. represented by the topographic wetness index TWI (Beven and Kirkby, 1979)). However, these proxies are still relatively coarse and typically lump the entire RZ into larger spatial units (e.g. model cells). Accordingly, small-scale heterogeneity of topography and hydrological properties, which can significantly affect the hydrologic connectivity of local source zones to the stream (Frei et al. 2010) are not adequately represented.

We argue that refined proxies that explicitly capture the smaller-scale heterogeneity of riparian zones could generally improve our mechanistic understanding of DOC exports from temperate catchments and potentially provide a means to infuse this understanding into DOC export models for larger scales. For example, Andersson and Nyberg (2009) proposed a skewed linear relationship between mean catchment-scale TWI and DOC concentration for various Swedish catchments, but it only performed well under wet conditions. We postulate that a riparian TWI, evaluated at a spatial resolution that resolves the small-scale topography within the RZ could improve the mechanistic basis of the DOC-TWI relationship proposed by Anderson and Nyberg (2009), as it would be better able to resolve the actual contributing source zones of DOC during different wetness states. DOC production will be highest in the wet depressions of the micro-topography within the RZ, which are defined by a high riparian TWI. Those depressions will also first be intercepted by a rising groundwater table during events, eventually leading to the development of surface flow networks (Frei et al. 2010), which episodically connect dominant DOC

source zones with the stream. We therefore hypothesize that both DOC production and transport to the stream are significantly controlled by the micro-topography of the RZ (lateral variability), and temporal dynamics of riparian groundwater level fluctuations (temporal variability). To test our hypothesis we characterized the micro-topography of a RZ using drone-based ortho-photography to obtain a digital elevation model (DEM) at high spatial resolution (1m). In addition, we monitored groundwater levels in a dense piezometer network, stream discharge, as well as DOC quantity and quality in stream water and groundwater. Stream flow generation and flow paths are modelled in detail with an integrated surface-subsurface numerical flow model, which explicitly accounts for micro-topography of the RZ. Explicit subsurface and surface flow paths from the model and DOC fingerprinting based on high-resolution mass spectrometry were used to identify dominant DOC source zones in the RZ and quantify different flow paths to the stream, which cause the observed space and time patterns of instream DOC concentrations.

## 2. Materials and Methods

### 2.1 Study site and site characterization

The study site is in a headwater catchment of the Rappbode stream (51°39'22.61"N 10°41'53.98"E, Fig. 1) located in the Harz Mountains, Central Germany. After draining into a drinking water reservoir, the Rappbode stream flows into the river Bode, and discharges (through the rivers Saale and the Elbe) into the North Sea. The catchment has an area of 2.58 km² and a drainage density of 2.91 km km⁻¹. The study site has a temperate climate (Kottek et al., 2006), with a long-term mean air temperature of 6.0 °C and mean annual precipitation of 831 mm (Stiege weather station 12 km away from the study site, data provided by the German Weather Service DWD). The uncultivated and uninhabited catchment is predominantly forested with spruce and pine trees (77 %), 11 % is covered with grass, and 12 % is covered by other vegetation and a few unpaved roads. Elevation ranges from 540 to 620 m above sea level; the mean topographic slope is 3.9°. The geology at this site consists mainly of graywacke, clay schist and diabase (Wollschläger et al., 2016). Soils in the spring area are dominated by peat and peat formation. Overall, 25 % of the catchment soils are humic and stagnic gleysols that are distinctive of riparian zones.

We flew a drone over the study site to measure the topography and used the data to create a digital elevation model (DEM) of the area with a spatial resolution of 1 m. Electric resistivity tomography (Resecs DC resistivity meter system, Kiel, Germany) was applied at two transects (Fig. 1b) using a Wenner alpha configuration with an electrode distance of 0.5 m in order to explore structural consistencies of the subsurface. The 90th percentile of the topographic wetness index (TWI) as a measure for the extent of riparian wetlands in the catchment (Musolff et al., 2018) is 10.10 (median 7.58). The study site was chosen to be within this percentile of the Rappbode catchment TWI (derived from the DEM) and is thus regarded to be representative for riparian sites of the catchment.

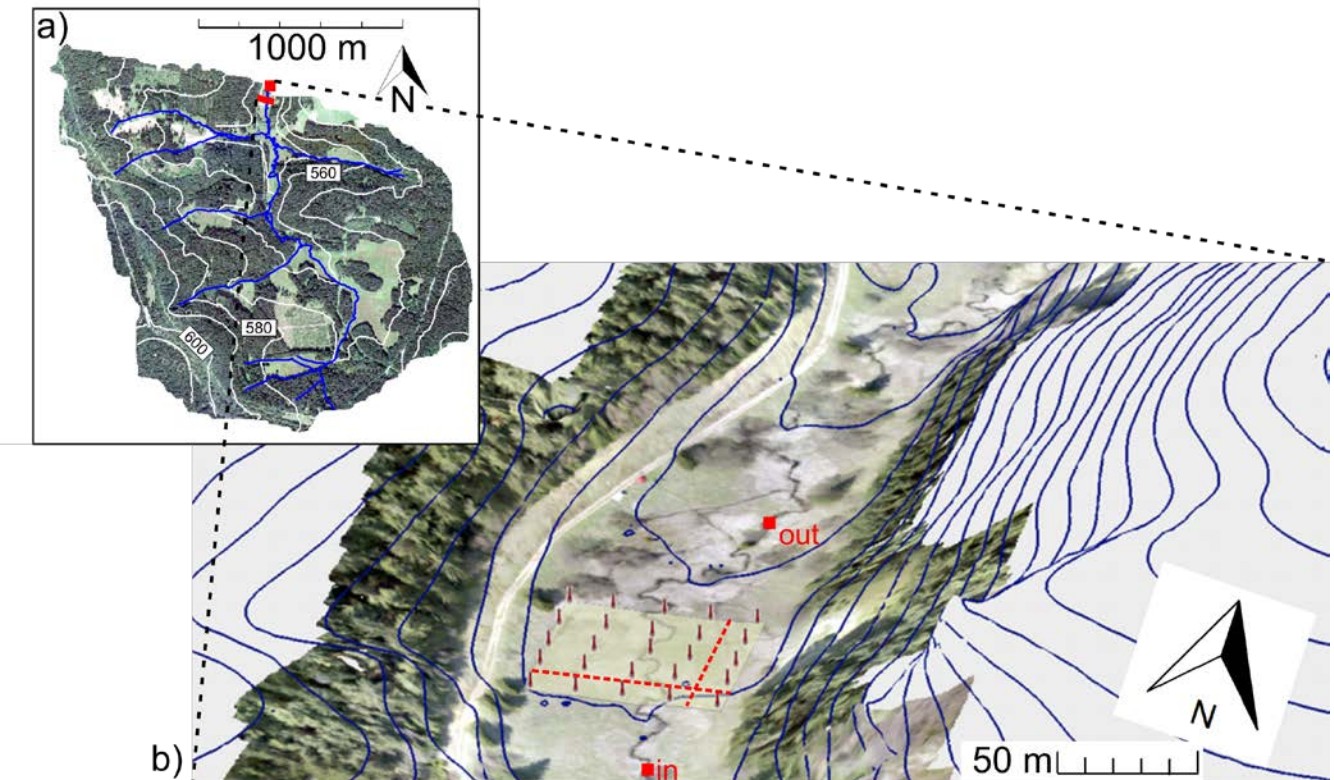

**Figure 1. a) Rappbode catchment and position of the study site (red square). b) 3D view of the study site (red squares indicate in and outlet stream gauges). Red pins indicate the soil sampling transect and the piezometer network. Location of electric resistivity tomography indicated by red dashed lines. Note the 2.5-fold vertical exaggeration. Map data: ©Google, GeoBasis-DE-BKG.**

### 2.2 The monitoring program

We carried out intensive field observations from 28 February 2017 until 19 December 2018, and continued the data collection
at a lower frequency until 23 July 2019. Note that within this campaign different probes cover partly different measurement periods due to a sequential deployment of the devices. Therefore we decided to set the period for the actual monitoring campaign for data analysis and modeling at 12 April 2017 to 19 December 2018, where the multiparametric dataset is most complete.

### 2.2.1 In-stream data sensors

Two PCM4 portable flow meters (Nivus, Germany) measured discharge in the Rappbode stream at a chosen inlet ($PCM4_{in}$, catchment area: 2.28 km² and outlet of the study site ($PCM4_{out}$, 2.54 km²), respectively. A pressure transducer (Solinst Levellogger, Canada) was installed in the center of the study site. All three probes measured water level every 15 minutes. Discharge at the center was then estimated via a stage-discharge relationship (stage was measured using a pressure transducer, $R^2 = 0.99$ (0.72 without the extreme value)) which was established based on biweekly manual discharge measurements using

an electromagnetic flow meter ($n$ = 37; MF pro, Ott, Germany). Maximum discharge measured manually was 0.22 m³ s⁻¹ on 28 February 2017 (the first measurement) at a water level of 37.9 cm. Manual measurements were recorded until 19 December 2018. The extrapolation of the stage discharge relationship to a wider range of stages was found to be in a valid range (Werner et al., 2019). However, values larger than 0.22 m³ s⁻¹ are more uncertain than smaller values.

### 2.2.2 Weather station

A weather station (WS-GP1, Delta-T, United Kingdom) was placed about 250 m northwest of the study site in order to characterize ambient weather conditions. Air temperature, humidity, wind direction and speed, solar radiation and rainfall were recorded at a 30 min intervals. Potential evapotranspiration ($ET_P$) was calculated from the weather data after Penman-Monteith (Allen et al., 1998) also at a 30 minute resolution.

### 2.2.3 Piezometer network

The DEM (Fig. 1b) revealed that the floodplain's slope in the direction of the stream (0.2m/10m) was steeper than the slope towards the stream (0.1m/10m). We expected that this would have ramifications for the direction of the slope of the groundwater level and its temporal dynamics. We installed a piezometer network aligned on a square grid, with one principal axis oriented in parallel to the stream and the other perpendicular to the stream to capture the temporal dynamics of the groundwater levels in both principal directions of this slope. We positioned 25 partly screened piezometers (2.54 cm diameter, HDPE, 10 cm filter length) in a rectangular grid pattern adjacent to the Rappbode stream, comprising a piezometer network covering 50 m × 50 m (Fig. 1b). The A horizon in the piezometer holes was 17.7 cm ± 2.4 cm on average ($n$ = 27) (Fig. 1). The well spacing was regularly 12.5 m in both principal directions of the grid, although installation depth was variable. The depth of the centers of the piezometer screens ranged between 20 cm and 107 cm below ground (average = 75.2 cm). This was a tradeoff between having continuous water level measurements from the pressure transducers and covering the anticipated large variety of different DOC characteristics in different soil layers and depths (Shen et al., 2015). We equipped each piezometer with a pressure transducer (Levellogger, Solinst, Canada and Diver, van Essen, Netherlands), measuring at a 15-minute interval. All pressure transducers were barometrically corrected and adjusted to manual measurements of the groundwater level at eight occasions during a 15 months measurement period (from 04 October 2017 to 19 December 2018). In addition we irregularly installed three wells with screens at 0.3 m depth (but no pressure transducers) inside the piezometer network for sampling near the surface. Fig. S1 gives an overview of the installed screen depth and the soil horizon accessed by the screened section.

### 2.2.4 Sampling and maintenance

At the monitoring site along the Rappbode stream, we overall collected 68 stream event samples during five events, 66 riparian samples and five stream samples during five occasions (one stream sample per occasion) and 38 routine stream samples (every two weeks), which were analyzed for DOC concentration. The molecular composition of the DOC was determined via Fourier-

transform ion cyclotron resonance mass spectrometry (FT-ICR-MS; see Table S1 for detailed information). Auto-samplers (6712 Full-Size Portable Sampler, Teledyne ISCO, US) were triggered by the rate of water level increase to sample stream water during discharge-generating events at least once per hour. Auto-sampler bottles (PP) were soaked for 48 h in 0.1 N HCl prior to use. We prepared process blanks with deionized water to correct for eventual contamination during field work and sample processing. Due to the remoteness of the study site, we collected auto-sampled stream water samples within 4 days after the triggered event sampling. Samples were stored in the dark inside the sampler and air temperature was always below 10°C during that time. We are aware that the delayed sample retrieval constitutes a limitation of our study which may affect DOC concentration and composition, in particular with respect to labile DOC sources, e.g. leaf leachate (Catalán et al. 2021). Yet, Werner et al. 2019 concluded that in-stream processing and biodegradation are likely to be of minor importance at our experimental site. Further, DOC composition typically shifts towards more stable, allochthonous DOC quality during events (Werner et al., 2019). Hence the major fraction of event-DOC is expected to be unaffected within the first four days (Mostovaya et al., 2016; Catalán et al., 2021). We collected riparian zone shallow groundwater samples from 3 to 18 out of the 28 installed piezometers depending on hydrological conditions during the five sampling dates. Generally groundwater sampling in summer turned out to be difficult due to low groundwater levels, which inhibited sampling of surface runoff and wells screened closer to the surface. To ensure a good comparability between sampling dates, we decided to focus on April and December samples, when a complete set of groundwater and surface water data was available. Before riparian sample collection, we replaced water in the wells one to three times (based on the responsivity of the wells) through pumping. We rinsed the flasks and the pump with sample water prior to sample collection and subsequently transferred 100 mL sample into acid-rinsed (0.1 N HCl) and baked (500 °C, 4 h) glass bottles. Additionally, stream water was collected at each of the five riparian sampling occasions. Samples were stored dark and cool until further processing in the laboratory.

### 2.3 Chemical analysis

### 2.3.1 Sample processing and DOC determination

Samples were filtered (0.45 µm membrane cellulose acetate filters, rinsed with 20 mL of sample water to avoid bleeding; Th. Geyer, Germany) and acidified to pH 2 (HCl, 30 %, Merk, Germany) on site. Subsequently samples were stored cool (4 °C) and dark until timely DOC measurement and extraction in the laboratory.

DOC concentration was determined as non-purgeable organic carbon with a high-temperature catalytic oxidation system (multi N/C 3100, Analytik Jena, Jena, Germany) from acidified samples and extracts after solvent evaporation. Due to the small difference in DOC concentration between hourly and subhourly samples during events we chose a lower time resolution for high resolution mass spectrometry measurements. A volume of 15 - 200 mL ($n = 142$) was extracted via solid-phase extraction (SPE) using an automated system (FreeStyle, LC Tech, Obertaufkirchen, Germany) on 50 mg styrene-divinyl-polymer type sorbens (Bond Elut PPL, Agilent Technologies, Santa Clara, CA, United States) to desalt the sample for subsequent direct infusion electrospray ionization mass spectrometry (DI-ESI-MS) according to Dittmar et al. (2008) and

(Raeke et al., 2017). The carbon-to-sorbens ratio (C:PPL) was $280 \pm 130$ (m/m, $n = 142$). The SPE-DOM (solid phase extractable dissolved organic matter) was eluted with 1 mL methanol (Biosolve, Valkenswaard, The Netherlands), and stored at $-20$ °C until measurement. Carbon based extraction efficiency was $(56 \pm 15)$ % (determined from $n = 133$ samples, Fig. S2). This is in the range of typical extraction efficiencies obtained for freshwater samples (Raeke et al., 2017). Immediately prior FT-ICR-MS analysis extracts were diluted to 20 ppm and mixed 1:1 (v/v) with ultrapure water (Milli-Q Integral 5, Merck, Darmstadt, Germany).

### 2.3.2 FT-ICR-MS measurement

An FT-ICR mass spectrometer equipped with a dynamically harmonized analyzer cell (solariX XR, Bruker Daltonics Inc., Billerica, MA, USA) and a 12 T refrigerated actively shielded superconducting magnet (Bruker Biospin, Wissembourg, France) instrument was used in ESI negative mode (capillary voltage: 4.2 kV) using an Apollo II source. Extracts were analyzed in random order with an auto sampler (infusion rate: 10 µL min$^{-1}$). For each spectrum, 256 scans were co-added in the mass range 150 - 1000 m/z with 4MW time domain (resolution @ 400 m/z was ca. 483000). Mass spectra were internally re-calibrated with a list of peaks (247 - 643 m/z, n > 55) commonly present in terrestrial DOM and the mass accuracy after linear calibration was better than 0.13 ppm ($n = 142$). Peaks were considered if the signal-to-noise (S/N) ratio was greater than four. Raw spectra were processed with Compass DataAnalysis 4.4 (Bruker Daltonics Inc., Billerica, MA, USA). Suwannee River Fulvic Acid (SRFA) reference sample and a pool sample (mix of randomly picked DOM extracts) was repeatedly measured to check instrument performance across multiple measurement days and solvent and extraction blanks were measured with the samples.

### 2.3.3 FT-ICR-MS data processing

Molecular formulas were assigned to peaks in the range 0-750 m/z allowing for elemental compositions $C_{1-60}H_{0-122}N_{0-2}O_{0-40}S_{0-1}$ with an error range of $\pm 0.5$ ppm according to Herzsprung et al. (2020). Briefly, the following rules were applied: $0.3 \leq H/C \leq 2.5$, $0 \leq O/C \leq 1$, $0 \leq N/C \leq 1.5$, $0 \leq DBE \leq 25$ (double bound equivalent, $DBE = 1 + 1/2$ (2C - H + N), Koch et al. (2014)), -$10 \leq DBE-O \leq 10$ (Herzsprung et al., 2014), and element probability rules proposed by Kind and Fiehn (2007). Isotopologue formulas ($^{13}$C, $^{34}$S) were used for quality control but removed from the final data set as they represent duplicate chemical information. All peaks present in the instrument blank and in the SPE blanks were subtracted from the mass list. Relative peak intensities (RI) were calculated based on the summed intensities of all assigned peaks in each sample. To ensure that the variance in DOC quality observed by FT-ICR-MS was not induced by systematic instrumental shifts at different times of the year, we quantified the variability of peak intensities based on 15 reference samples (SRFA) of the four measurement days. Subsequently this variability was applied to every RI in measured samples to derive a mean error for the intensity (see S1). We conclude that the analytical uncertainty from the FT-ICR-MS measurements between the different measurement dates does only minor affect the overall variance of the samples, which allows the joint evaluation of all samples (see S1, Fig. S3, Fig.

S4). An assigned molecular formula is termed compound throughout this article although they potentially represents multiple
isomers.

## 2.4 Numerical water flow modeling

The numerical code HydroGeoSphere (HGS) was used to quantify water flow at the study site. HydroGeoSphere is a 3D
numerical model describing fully coupled surface-subsurface, variably saturated flow (Therrien et al., 2010). It solves
Richards' equation for 3D variably saturated water flow in the subsurface domain, and uses Manning's equation and the
245 diffusive-wave approximation of the St. Venant equations to simulate surface flow in the 2D surface domain and 1D channel
network (Yang et al., 2015). Using a dual node coupling approach, HydroGeoSphere simulates the water exchange fluxes
between the domains, providing the simulated infiltration/exfiltration fluxes. More details on the governing equations,
coupling approach, and general aspects of HydroGeoSphere can be found in Therrien et al. (2010).

Only the upper 2 m of the alluvial sediments were included in the flow simulation as an aquifer because geological survey
data showed that the electric resistivity dropped sharply below that depth indicating the presence of bed rock (Fig. S5). The
subsurface was discretized into 8 horizontal element-layers, each composed of 6924 prisms. The layer thicknesses ranged
from 0.05 m near the land surface to 0.5 m near the aquifer bottom. The horizontal cell sizes varied from 1 m to 2 m. The
6924 uppermost 2D triangles of the 3D prismatic mesh were used to discretize the surface domain. The channel crossing the
study site was discretized into 148 1D segments, which coincide with the segments of the 2D triangular mesh. The line element
made up of the channel segments was treated as a Cauchy boundary with the stream stage being calculated based on the
assumption of a rectangular cross-section with channel width and depth based on measurements.

### 2.4.1 Parameters

Horizontal variability of the saturated hydraulic conductivity $K$ was calibrated using 38 pilot-points (Tang et al., 2017; Moeck
et al., 2015) distributed inside the study site. Each of these pilot-points were associated with a $K$ value, set to 0.1 m d$^{-1}$ prior
to calibration. For the vertical $K$ heterogeneity, it was assumed that $K$ was depth-dependent and decreased exponentially when
the aquifer was deeper than 0.2 m, as $K = K_0$ when $d < 0.2$ m, and $K = K_0 e^{-\lambda d}$ $d > 0.2$ m, where $K_0$ is the hydraulic conductivity
of the aquifer top determined from the horizontal $K$ field, $d$ is the depth below the land surface and $\lambda$ is a factor constraining
the decreasing rate, set to 0 prior to calibration. These formulations captured the general decreasing trend of $K$ with depth,
while also reflecting the fact that this decreasing trend was not significant in the upper 0.2 m of the soil, which contained most
roots and mainly consisted of poorly decayed organic material.

The surface domain and channel domain were uniformly parameterized with Manning roughness coefficients (Manning et al.,
1890), respectively. Prior to calibration, the roughness coefficients were set to 6·10$^{-6}$ d m$^{-1/3}$, a typical value for
floodplains/grassland. The parameters described above were selected as key parameters that could significantly influence the
flow processes, and were optimized during calibration (Table S2). Other parameters were assigned for the model domain

according to literature values from Yang et al. (2018) from a nearby (25 km) catchment with similar geological settings. Values were then adjusted during calibration (Table S2).

### 2.4.2 Boundary and initial conditions

Input data was defined at a one-hour time resolution for the simulation, and all higher-resolution data (15 min) was aggregated accordingly. For the aquifer top boundary, spatially uniform and temporally variable precipitation was applied to the surface
domain. Spatially uniform and temporally variable potential ET, estimated using the climate data, was specified as model input with actual ET being simulated by the model (Therrien et al., 2010). For the upstream boundary AB (Fig. 2), a constant groundwater head gradient of 0.02 in the direction of the stream was assumed according to the measured groundwater levels, such that a groundwater flux ($Q_{up}$) entering the subsurface domain across AB could be determined using Darcy's law. A temporally variable flux $Q_{up}^c$ was directly applied to the inlet of the channel domain, representing the measured channel
discharge rate. The groundwater recharge rates via the two lateral boundaries of the model domain AD and BC ($Q_{left}$, $Q_{right}$) were estimated using $R \cdot A_{con}$, where $R$ is the annual mean groundwater recharge rate in this area (~200 mm yr$^{-1}$), and $A_{con}$ is the contributing surface area associated with each lateral boundary estimated from the DEM. The respective recharge fluxes $Q_{left}$ and $Q_{right}$ were calculated as 0.18 m$^3$ s$^{-1}$ per unit length and 0.09 m$^3$ s$^{-1}$ per unit length. They were also allowed to vary by 0.1 to 10 times of their initial values during model calibration (Table S2). Water can exit the model domains through the
downstream boundary CD, either via the subsurface calculated using a constant groundwater head gradient of 0.02 ($Q_{down}$), or via the surface domain ($Q_{down}^o$) and channel outlet ($Q_{down}^c$) calculated using a critical depth boundary condition (Therrien et al., 2010). All other model boundaries were assumed to be impermeable (no flow boundaries).

A steady state model was obtained by running a preliminary simulation using time-invariant boundary conditions. The steady state results were used as initial conditions for the actual transient simulations to reduce the influence from inappropriate
initial conditions.

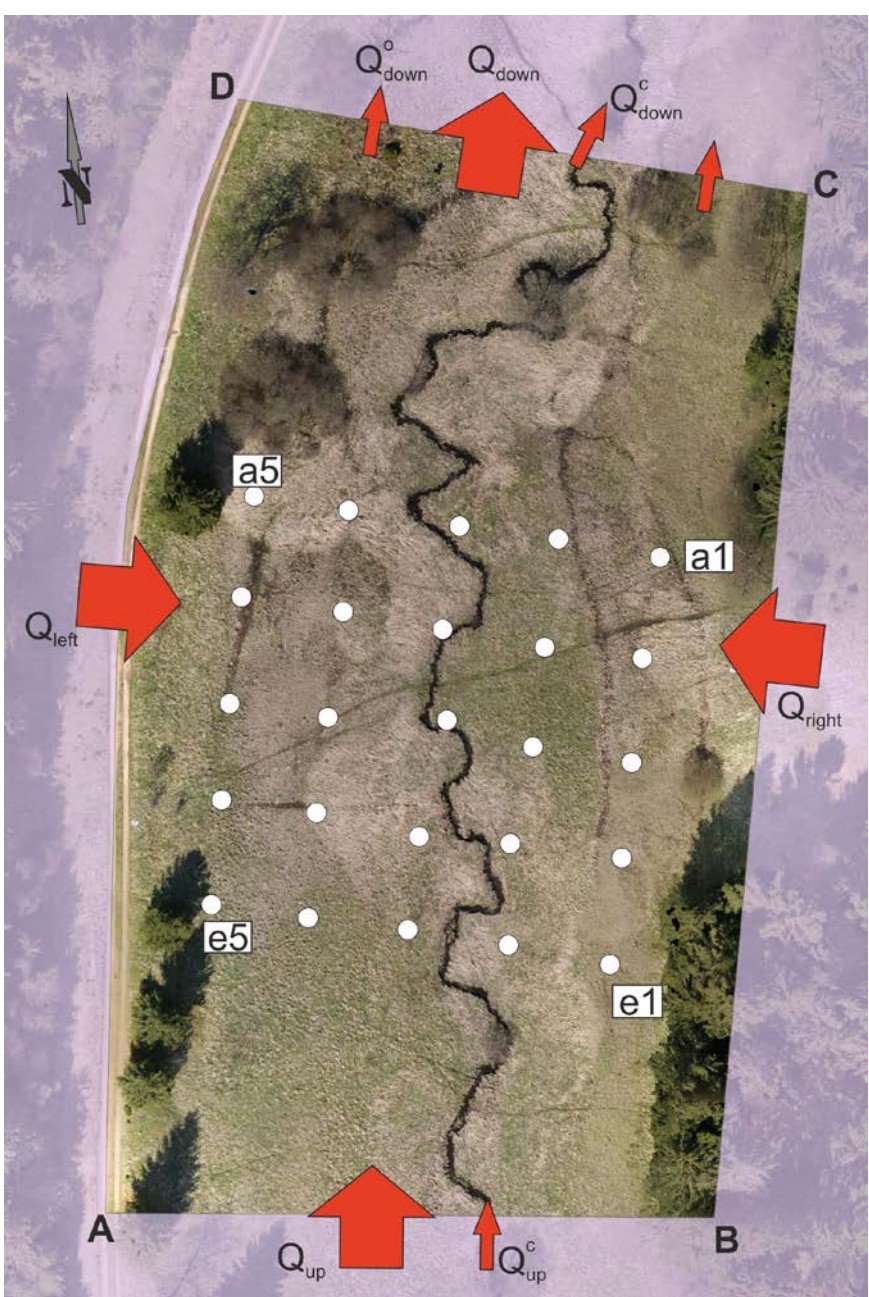

**Figure 2. The boundary conditions of the study site (indicated by polygon A-D).** $Q_{up}$: groundwater influx (const.), $Q_{up}^c$: channel influx (dynamic), $Q_{down}$: groundwater leaving the model site (dynamic), $Q_{down}^c$ and $Q_{down}^o$: surface water leaving the model site through channel outlet and through CD (dynamic), $Q_{left}$ and $Q_{right}$: groundwater discharge rate from side boundaries (const.). Red arrows indicate flow direction of the water. The 25 white points indicate wells which were used for model calibration. Labels of the piezometer run from a to e (lines) and 1 to 5 (columns) respectively.

### 2.4.3 Calibration

Transient calibration was performed using the software package PEST, which uses the Marquardt method to minimize a target
function (describing the error between modeled and measured variables; see S2 for the objective function) by varying the
values of a given set of parameters until the optimization criterion is reached (Doherty and Hunt, 2010). The calibrated model
parameters (Table S2) were set to be adjustable within the selected ranges around their initial values. The measured
groundwater level time-series at 25 observation wells and channel flux time-series at the outlet (similar location to $Q_{down}^c$)
were used to compare with the simulated ones (Fig. S6, such that the target function could be calculated. Because two different
data sets (groundwater level and channel flux) were used, a weighting scheme was selected to let the defined multi-objective
function be dominated by the data set of groundwater levels, because: (i) we focused more on the groundwater flow and the
associated surface-subsurface exchange fluxes, and (ii) the channel flux was relatively easy to reproduce by the model as
measured channel fluxes were directly assigned to the channel inlet. A 21-day period (15 November 2017 to 6 December
2017) was selected for the model calibration in view of the high CPU time demand for transient model runs, data availability
constraints and data variability requirements. Wet and intermediate conditions generate almost all of the runoff of the riparian
zone. The selected calibration period thus incorporates fluctuations during intermediate and high groundwater situations and
therefore covers both system states (from subsurface- to surface-flux dominated).

The time-variable groundwater levels were well replicated by the model for the wells near the channel (Fig. S6a). The wells
close to channel had better fits than those near the side boundaries (Fig. S6b), because the latter were more strongly constrained
by the constant groundwater fluxes through the side boundaries. The calibrated flow model was used to quantify internal water
flux data from specific regions in the riparian zone. Additionally, advective-dispersive particle-tracking was used on the flow
field from the calibrated model to visualize surface and subsurface flow paths through the model domain. The surface flow
paths are used to identify key runoff generation zones in the riparian zone.

### 2.5 Statistical methods

Statistical analysis was performed using R (R-Core-Team, 2017). Evaluation of geospatial properties was conducted via R in
combination with ArcMap (ESRI, US).

### 2.5.1 Chemical classification of potential DOC source zones

Peak intensity weighted average (*wa*) of FT-ICR-MS derived molecular parameters (mass to charge ratio (mz), elemental ratios
(H/C, O/C, N/C, S/C), nominal oxidation state of carbon (NOSC) and aromaticity index (AI)) was calculated for each sample
by Eq. (1):

$$wa_p(x) = \frac{\sum p_i(x) \cdot int_i(x)}{\sum int_i(x)},$$ 
(1)

where $wa_p(x)$ is the weighted average value for the molecular parameter $p$ in sample $x$. $p_i(x)$ is the derived value for the parameter $p$ of each molecular formula $i$ in sample $x$. Accordingly, $int_i(x)$ is the peak intensity for molecular formula $i$ in sample $x$.

A principal components analysis (PCA) was then performed with the riparian samples ($n = 66$) using the FT-ICR-MS derived $wa$ molecular parameters of all molecular formulas commonly detected in all investigated samples of the monitoring site (i.e. including the event samples, $n = 68$, and the base flow samples, $n = 5$; Table S1) covering on average 40 % of the assigned intensities in each sample. A consecutive k-means clustering on the first two principal components (R package FactoMineR (Lê et al., 2008)) was used to partition the riparian samples into two (as suggested by the silhouette index (Rousseeuw, 1987))

chemically distinct clusters, representing different DOC quality in the riparian groundwater. The Wilcoxon rank sum and the Kolmogorov Smirnov (KS) test were applied to identify significant differences in the distributions and medians of DOC concentration and FT-ICR-MS derived molecular parameters for the two groundwater DOC clusters and stream water samples.

### 2.5.2 Hydromorphological classification of potential DOC source zones

Every groundwater level time series was correlated (Pearson's r) with the stream water level time series. Geomorphological
analysis was conducted via the TWI, according to Eq. (2)

$$TWI = \log\left(\frac{f}{\tan(s)}\right), \tag{2}$$

Where TWI is the topographic wetness index for each cell, $f$ is the flow accumulation (the accumulated number of all cells topographically draining into a downslope cell) at each cell and s is the slope in radians of respective triangular surface element. We applied the DInf algorithm to calculate a realistic hydrological routing (Tarboton, 1997). The DInf algorithm
determines flow direction as the steepest downward slope on eight triangular facets formed in a 3x3 cell window centered on the cell of interest. To account for mathematical infinity/indefinite terms, zero slopes were set to 0.001 rad, and cells with no flow accumulation ($f = 0$) were set to 1 cell instead.

A smoothed map of the local $TWI_{HR}$ values was created by assigning the median TWI value of the central cell and its 8 surrounding cells to the central cell. According to KS and F-test statistics, the resulting map represented the non-smoothened
TWI distribution of the study site ($p_{KS} = 0.33$; $p_F = 0.76$). We applied the Wilcoxon rank sum to test for differences in $TWI_{HR}$ distributions and medians of the two DOC clusters. The median TWI value of the $DOC_I$ cluster was used as a manually chosen threshold to separate the RZ into two explicit zones of high and low TWI values. The water balance for the entire model site and the two TWI-generated zones was then estimated and compared to each other between 12 April 2017 and 19 December 2018 by modeling with HydroGeoSphere.

## 3 Results

### 3.1 Hydroclimatic conditions and DOC chemical characterization

A summary of the statistics of discharge, groundwater level and climatic variables throughout the 15-months measurement period is presented in Table 1. Discharge showed high variability at the event-scale. At annual scale, discharge expressed a clear seasonal pattern, with lowest values in late summer and highest values in spring (Fig. 3a). Stream water level was highest during a flood event from 01 to 03 January 2018 when the Rappbode stream went over-bank. We decided to not include this event in the statistics, because we could not estimate the discharge for water levels higher than the stream banks. Yet observing this flood event helped to verify and understand riparian surface runoff pathways at our study site. The amount of precipitation during 2018 (580 mm) was below the long-term annual mean (831 mm) at the nearest official weather station. Air temperature exhibited a seasonal pattern and was above the long term annual mean at the nearest station (8.6°C vs. 6.0°C).

**Table 1: Summary statistics of climatic, hydrological and chemical parameters of the study site and samples collected during the monitoring campaign between 12 April 2017 and 19 December 2018. $ET_0$: potential evapotranspiration, DOC conc.: DOC concentration, 'wa' indicates peak intensity weighted average values of the FT-ICR-MS derived molecular parameters: $wa_{mz}$ (mass to charge ratio), $wa_{HC}$ (hydrogen to carbon ratio), $wa_{OC}$ (oxygen to carbon ratio), $wa_{SC}$ (sulfur to carbon ratio), $wa_{AI}$ (aromaticity index), $wa_{NOSC}$ (nominal oxidation state of carbon).**

| | mean | sd | min | max |
|---|---|---|---|---|
| Air temperature [°C] | 8.6 | 8.18 | -18.9 | 34 |
| Rain [mm $h^{-1}$] | 0.11 | 0.62 | 0 | 31.8 |
| $ET_0$ [mm $d^{-1}$] | 1.65 | 1.24 | 0 | 4.6 |
| Stream water level [cm] | 14.64 | 8.48 | 3.54 | 69.94 |
| Discharge [L $s^{-1}$] | 58.7 | 92.9 | 8.2 | 1116.0 |
| DOC conc. [mg $L^{-1}$][1] | 3.80 | 2.77 | 0.69 | 15.77 |
| $wa_{mz}$ [1] | 436 | 8 | 420 | 453 |
| $wa_{HC}$ [1] | 1.28 | 0.05 | 1.15 | 1.41 |
| $wa_{OC}$ [1] | 0.40 | 0.01 | 0.36 | 0.45 |
| $wa_{SC}$ [x$10^3$][1] | 8.6 | 2.2 | 4.9 | 15.8 |
| $wa_{AI}$ [1] | 0.09 | 0.02 | 0.05 | 0.15 |
| $wa_{NOSC}$ [1] | -0.41 | 0.06 | -0.55 | -0.16 |

[1] for groundwater and riparian surface water samples taken from April 2018 to July 2019 (single spots were sampled multiple times throughout year, $n = 66$). A breakdown of average molecular parameters into the DOC clusters and the corresponding stream and event data can be found in Table S3.

Stream water levels (Fig. 3c) were closely coupled with groundwater levels, with lower and more fluctuating water levels in summer and less variable, higher water levels in winter (Fig. 3a). Water level fluctuations in wells closer to the stream followed

stream stage variations more closely than in the wells more distant to the stream, which showed more damped dynamics. This results in Spearman correlations ($r_s$) of groundwater level time series with the stream between 0.43 and 0.86 (mean of all $r_s$ = 0.60). The groundwater table was shallow throughout the measurement period with highest values in winter and after
snowmelt in spring (Fig. 3b).

An overview of hydroclimatic data for the dates of DOC sampling (Fig. 3b, c) in the stream or the RZ is given in Table S4. DOC concentrations in the stream during events generally followed the hydrograph, with higher concentrations during higher water levels. DOC concentration and molecular properties across all riparian groundwater samples exhibited a distinct variability (Table 1). In general, DOC in riparian water samples was of unsaturated and phenolic composition ($wa_{HC}$ = 1.27 ±
0.05; $wa_{OC}$ = 0.40 ± 0.01; $n$ = 66), that is typically found in wetland surface soils (LaCroix et al., 2019). Stream event samples significantly differed ($p < 0.001$) from riparian samples and were more unsaturated ($wa_{HC}$ = 1.17 ± 0.05; $n$ = 76) and more oxygenated ($wa_{OC}$ = 0.43 ± 0.03) as shown in Fig. S7.

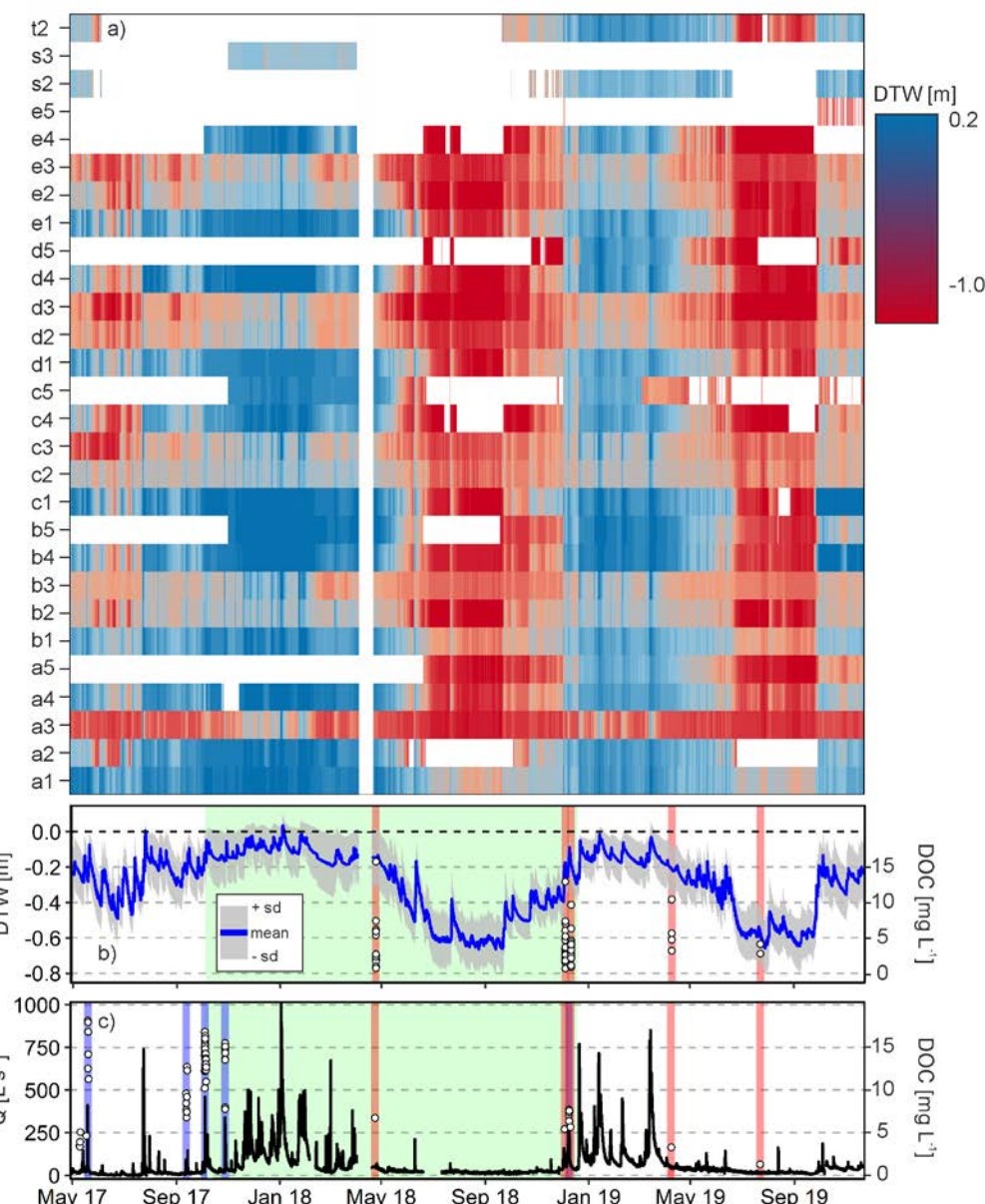


Figure 3. a) Depth to water table (DTW) time series of all 28 riparian wells. White spaces indicate missing data in the time series: due to varying (de)installation times (if white in the beginning), sensor failure (white gaps in every time series) and dryness-induced disconnection of wells with the groundwater (white gaps in red areas). Locations of wells are depicted in Fig. 2. b) Time series of mean DTW values (blue line) ± the standard deviation (grey ribbon). A positive value of DTW indicates water ponding at the soil

surface or that the piezometer measured a locally confined part of the aquifer. Red vertical bars indicate sampling dates of riparian groundwater. Points within the red bars show DOC concentration of respective riparian samples. Green area indicates the modeling period where ground water levels were used for calibration. c) Discharge time series of the central pressure transducer. Blue vertical bars indicate stream water auto sampling dates during events, red vertical bars indicate grab sampling dates of stream water. Points within blue and red bars show DOC concentrations of according stream water samples. Green area indicates the modeling period

where ground water levels were used for calibration. Note that the flood event at the beginning of January 2018 is ungauged.

## 3.2 Classification and mapping of potential DOC sources

### 3.2.1 Chemical classification

A detailed overview of the FT-ICR-MS results of the distinct water samples can be found in S3, Tables S3&S5, and Figs. S8-S10. The PCA to classify riparian DOC quality was able to explain 66.3 % of the total variance of FT-ICR-MS peak intensities using two principal components (PCs). K-means clustering based on the PCs then separated the riparian samples into two clusters of 19 and 47 samples ($DOC_I$ and $DOC_{II}$, resp.; Fig. S11), representing distinct DOC quality in the riparian zone. Weighted average molecular parameters were significantly different between the two clusters, allowing for a clear separation between $DOC_I$ and $DOC_{II}$ (Fig. 4a, Table S3). In contrast to the $DOC_{II}$ cluster, samples belonging to the $DOC_I$ cluster had higher DOC concentration and their molecular composition was characterized by more oxidized (higher NOSC and $wa_{OC}$), more aromatic molecules (higher $wa_{AI}$), with a lower fraction of heteroatoms (smaller $wa_{SC}$, $wa_{NC}$ not shown), and a lower molecular weight (smaller $wa_{mz}$). Regarding the high resolution event sampling, all samples were used, but inter-event variance of DOC properties was higher than intra-event variance. Therefore we considered bulk sample properties of every event to be satisfactory for comparison with the riparian samples (Fig. 4a). Comparison of $DOC_I$ and $DOC_{II}$ molecular parameters and concentration with that of stream water sampled during rain events in spring, summer and autumn confirmed overall different DOC quality between riparian groundwater and stream water (based on median values and ranges; Fig. 4). However, median values of the $DOC_I$ cluster were always closer to the median of stream water event samples than the respective $DOC_{II}$ median. Moreover, the DOC molecular parameters of one event in December (Fig. 4a, orange dots) was in the range of the riparian samples, but did not show much compositional variability within the event.

$DOC_I$ samples from April ($n = 9$, Fig. 3) and December ($n = 9$) did not show significant differences in DOC molecular parameters (except $wa_{HC}$) and concentration (Fig. 4b). In addition, DOC concentration and quality in the stream samples (from the routine measurement program, non-event conditions) generally matched $DOC_I$ concentration and quality in April and December (except $wa_{HC}$ and $wa_{AI}$). In contrast, $DOC_{II}$ samples from April ($n = 13$) and December ($n = 33$) differed significantly according to their $wa_{mz}$, $wa_{HC}$, $wa_{AI}$, $wa_{NOSC}$ values and DOC concentration (Fig. 4c). While DOC concentration and quality of stream water samples from December were mostly within the range of the respective $DOC_{II}$ samples, stream water samples from April were mostly outside the range of the DOC molecular parameters and concentrations of the respective $DOC_{II}$ samples.

Both DOC clusters were associated with groundwater sampled at depth to water table (DTW) > -0.3 m in eight cases. Median high-resolution TWI ($TWI_{HR}$) values at the well position (see 2.5) were grouped according to their attribution to the $DOC_I$ and $DOC_{II}$ clusters based on the chemical characterization. Note that 7 (i.e. 6 wells and one surface pond sample) out of 15 locations occur in both DOC clusters as DOC quality varied over time. According $TWI_{HR}$ values also contribute to both clusters (Fig. 4d). In general, the median values of $TWI_{HR}$ for wells attributed to samples of the $DOC_I$ cluster were significantly higher (Wilcoxon rank sum $p < 0.008$) than respective values of the samples of $DOC_{II}$ cluster (Fig. 4d). The distribution

(median) of TWI$_{HR}$ was different (higher) when comparing the TWI$_{HR}$ values of DOC$_I$ vs DOC$_{II}$ samples from April, whereas December samples did not show any statistical significant difference in their TWI$_{HR}$ distribution or median.


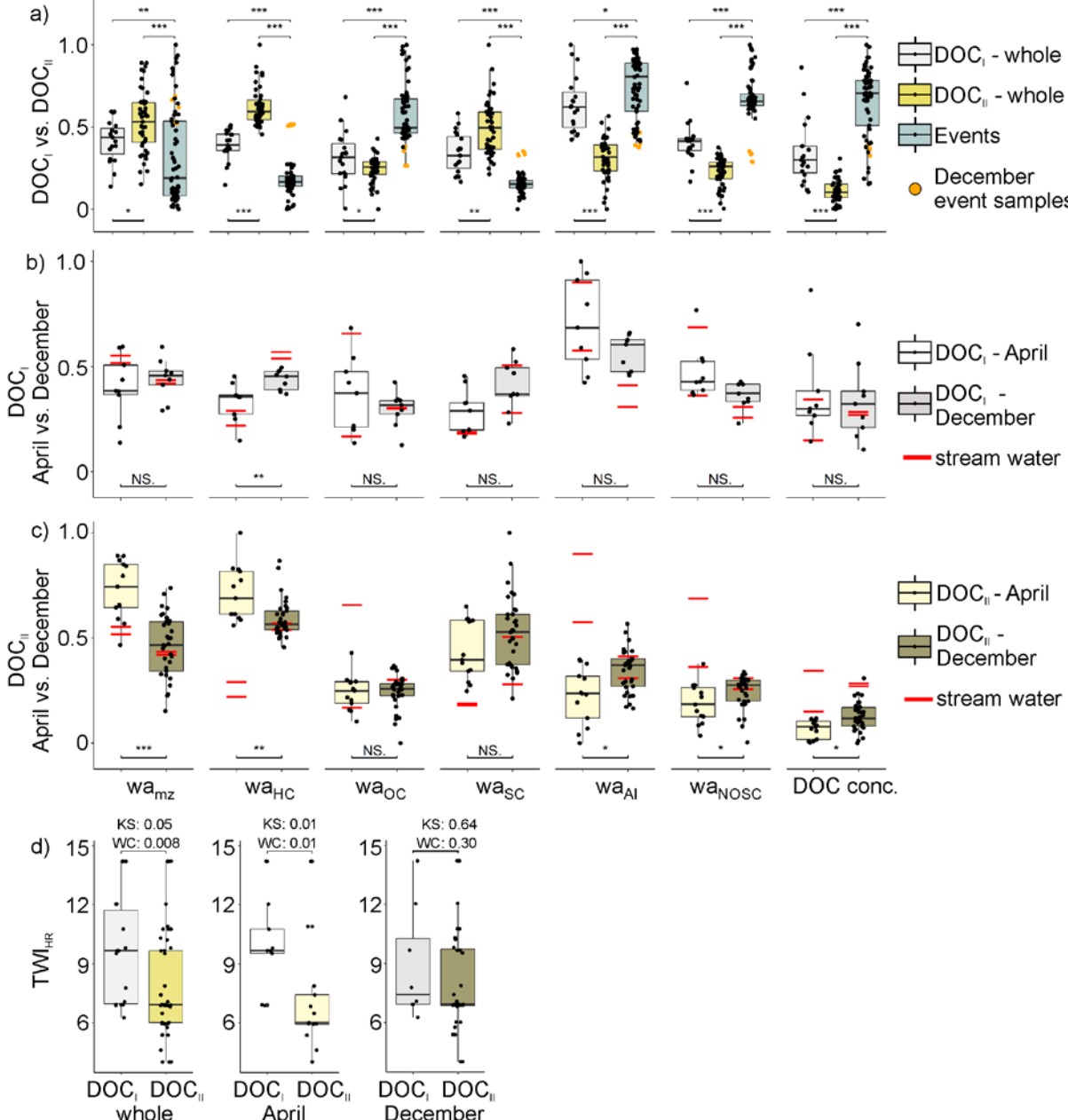

**Figure 4.** Comparison of FT-ICR-MS derived weighted average ('*wa*') molecular parameters $wa_{mz}$ (mass to charge ratio), $wa_{HC}$ (hydrogen to carbon ratio), $wa_{OC}$ (oxygen to carbon ratio), $wa_{SC}$ (sulfur to carbon ratio), $wa_{AI}$ (aromaticity index) and $wa_{NOSC}$ (nominal oxidation state of carbon) as well as DOC concentration (DOC conc.) and high-resolution TWI ($TWI_{HR}$) values of stream, $DOC_I$ and $DOC_{II}$ samples. a) Boxplots of molecular parameters and DOC concentration of the $DOC_I$ and $DOC_{II}$ clusters. Orange dots indicate samples of one December event. b) Boxplots of molecular parameters and DOC concentration of April and December $DOC_I$ samples and all event samples. c) Boxplots of molecular parameters and DOC concentration of April and December $DOC_{II}$ samples and all event samples. Red horizontal lines in b) and c) indicate weighted averages of two April and two December stream water samples collected during the respective riparian groundwater sampling campaign. Data in a) to c) were min-max normalized to values between 0 and 1 for better illustration (see Table 1 for actual values). d) Boxplots of $TWI_{HR}$ values as affiliated to the respective wells of $DOC_I$ and $DOC_{II}$ samples. Wilcoxon rank sum (WC) and Kolmogorov Smirnov (KS) test results are depicted above the squared brackets. Squared brackets above and below boxplots in a) to c) indicate the application of a KS test between two partitions. Asterisks indicate p-values of the KS test (∗∗∗: < 0.001; ∗∗: < 0.01; ∗: < 0.05; NS: not significant).

### 3.2.2 Spatial mapping

The significant difference in median $TWI_{HR}$ values of well locations contributing to the $DOC_I$ and $DOC_{II}$ clusters (Wilcoxon rank sum $p < 0.008$) was used to spatially separate potential source zones by using the median $TWI_{HR}$ value of the well locations of the $DOC_I$ cluster (9.66) as a threshold. Using this manually chosen threshold allowed to allocate the samples of both DOC clusters to two distinct $TWI_{HR}$-based groups. In this way, more than 50% of samples contributing to the $DOC_I$ cluster constitute one group while allowing less than 25% (15% in April) of the samples contributing to the $DOC_{II}$ cluster in that group (again

note that 7 out of overall 15 sampling locations appear in both TWI groups). Extending this $TWI_{HR}$-based grouping to the entire study site, the riparian zone was divided into zones of high $TWI_{HR}$ ($DOC_I$ source zone in the following) and low $TWI_{HR}$ ($DOC_{II}$ source zone) values (Fig. 5). The high $TWI_{HR}$ zones defined in this way represent 14.6 % of the area of the study site.
 The HydroGeoSphere (HGS) model was then used to quantify the runoff generation from the delineated $DOC_I$ source zones and to quantify their impact on total runoff generation and DOC export from the study site. According to our simulations

surface runoff, which we define here as all water running off at the surface eventually reaching the stream (originating from groundwater exfiltrating to the land surface or direct precipitation onto saturated areas), was the main source of total flow gain in the stream over the simulation period (Fig. 6). Surface inflows into the channel constituted 66 % of the total flow gain along the simulated stream segment over the simulation period. The median contribution of surficial runoff to total runoff generation during the model period was 61 % (± 12 % SD) but surface contributions increased up to 99 % during event situations. We

selected the subsurface-surface exchange flux as a key descriptive variable for potential surface runoff contributions, because it quantifies the availability of water at the surface for each cell of the model. Although there was a 1.5 times higher net surface water flux generation from low $TWI_{HR}$ zones throughout the modeling period (Fig. 6b), the median of the area-normalized water exchange flux for $DOC_I$ source zone (high $TWI_{HR}$ zones; 0.026 m d$^{-1}$) was about 8.6 times higher than that for $DOC_{II}$ source zones (low $TWI_{HR}$ zones; 0.003 m d$^{-1}$). This resulted in higher absolute exchange fluxes in high $TWI_{HR}$ zones in about

47 % of the modeling period. During (non-winter) runoff events, water exchange flux contribution of high $TWI_{HR}$ zones increased up to 100 % (negative or no exchange flux for $DOC_{II}$ source zones in dry summer). Low $TWI_{HR}$ zones contributed more potential surface runoff at non-event winter conditions and flooding events when high overall exchange fluxes occurred under fully saturated soil conditions. Hydrological conditions were exemplary mapped for situations on 13 December 2017,

immediately after an event under wet antecedent conditions, and on 29 August 2018, amidst a prolonged dry period in summer

(Fig. S8, Table S6 for corresponding water fluxes). High $TWI_{HR}$ zones had the highest exchange fluxes and water depths in both wet and dry situations. Surface flow paths in winter intersect the high $TWI_{HR}$ zones establishing hydrologic connectivity between these zones and the stream, which is in line with our observations on DOC quality. The highest positive exchange flux values (GW exfiltration) occurred at the outer hillslope boundaries of the RZ, running parallel to the channel (values at the exact boundaries of the model have not been taken into account due to potential boundary effects). These exfiltration spots

were located close to the strongest surface water infiltration spots. During the exemplary wet situation, the entire RZ was saturated with water besides the stream banks. Surficial runoff pathways then connect the $DOC_I$ source areas (high $TWI_{HR}$ zones), running parallel to the stream and eventually entering it within the modeled domain.

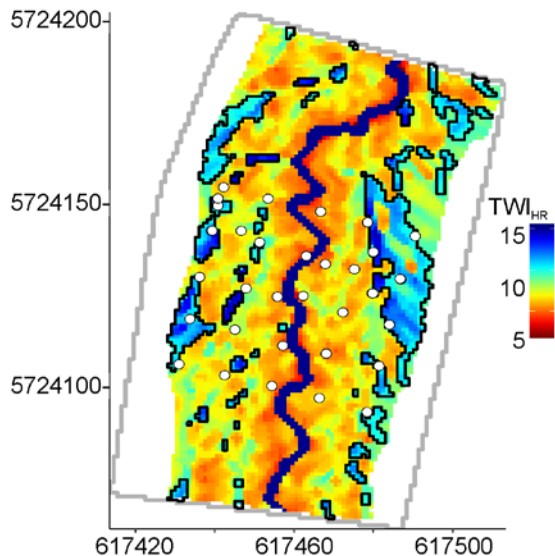

**Figure 5. High resolution TWI ($TWI_{HR}$) map of the modeled site (excluding hillslopes). White points indicate sampling locations, the**
**Rappbode stream is indicated by the blue line. Black polygons are high $TWI_{HR}$ zones, indicating $DOC_I$ source areas.**

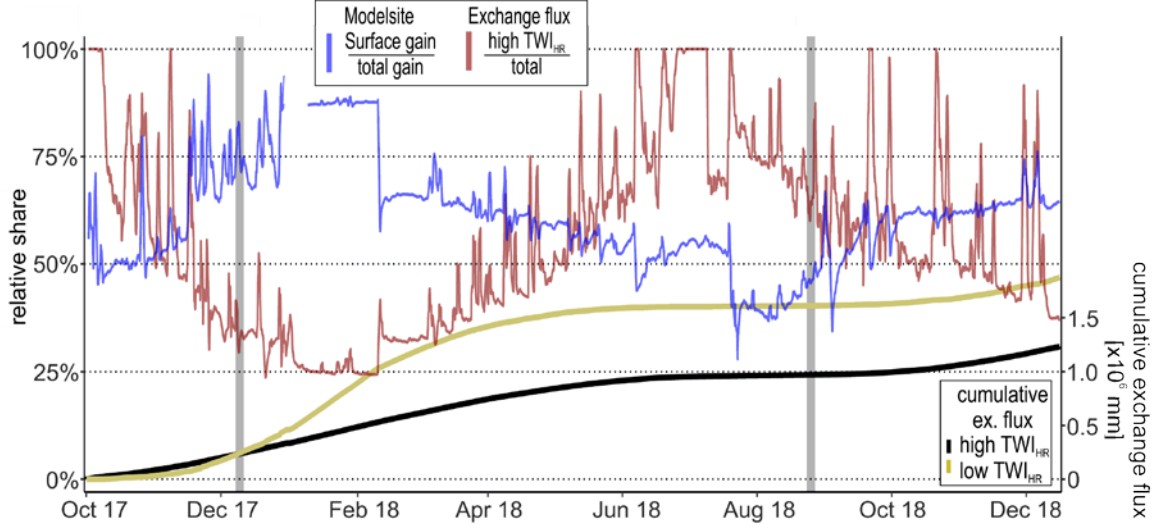

**Figure 6. Share of surface runoff on total runoff generation (blue line) and of high-TWI$_{HR}$ zone water exchange flux on total exchange flux in the model site (red). Data was smoothed to daily values for better visualization. The gap in the blue graph in January 2018 is due to an ungauged flood event. Cumulative positive water exchange flux of high TWI$_{HR}$ (black) and low TWI$_{HR}$ (yellow/khaki) source zones shown on second y axis. Grey bars indicate modeling dates for a wet situation on 13 December 2017, right after an event and at dry conditions and on 29 August 2018, amidst of a longer dry period in summer (Fig. S8, Table S6).**

### 3.3 Surface DOC export from high TWI$_{HR}$ and low TWI$_{HR}$ source zones

During the model period, wells in DOC$_I$ source zones had a median DOC concentration of 5.8 mg L$^{-1}$ (mean ± SD: 6.2 ± 2.7 mg L$^{-1}$), which was 2.3 times higher than the median for the DOC$_{II}$ source wells (2.7 ± 1.2 mg L$^{-1}$). We assumed the DOC concentrations to stay in a range of mean ± SD throughout the year (cf. Fig. 4b, c). DOC export was then roughly calculated by multiplying mean ± SD of DOC$_I$ and DOC$_{II}$ concentrations with the absolute surface runoff volumes from the respective high and low TWI$_{HR}$ zones. With that mean overall export from high TWI$_{HR}$ zones exceeded that from low TWI$_{HR}$ zones in about 70 % of the time although making up only 14.6 % of the total area. In absolute numbers, high TWI$_{HR}$ zones exported roughly 1.5 times the amount of DOC (7.1·10$^6$ g) to the stream than low TWI$_{HR}$ zones (4.6·10$^6$ g). This amounts to a nearly 20 times higher area-normalized DOC export from high TWI$_{HR}$ zones than from low TWI$_{HR}$ zones. Highest disparity between the export of the two source zones was during events in autumn and spring when surface water in the low TWI$_{HR}$ zone infiltrated instead of rapidly flowing into the stream (no DOC export from low TWI$_{HR}$ zones, Fig. 7) while high TWI$_{HR}$ zones exported DOC (positive spikes). Infiltrating conditions for the high TWI$_{HR}$ zone only occurred during summer events when DOC export was generally at the minimum (mean daily export rates of 3.1 and 17.3 g d$^{-1}$ for low and high TWI$_{HR}$ zones, respectively) whereas equally high DOC export occurred in winter (234.2 g d$^{-1}$ and 267.2 g d$^{-1}$ for low and high TWI$_{HR}$ zones, respectively). The median export from the high TWI$_{HR}$ zone was above that from low TWI$_{HR}$ zone in non-winter conditions, with the highest disparity between medians in spring and autumn (Fig. 7). Then high TWI$_{HR}$ zones exhibited exfiltrating conditions, whereas water in low TWI$_{HR}$ zones kept infiltrating.

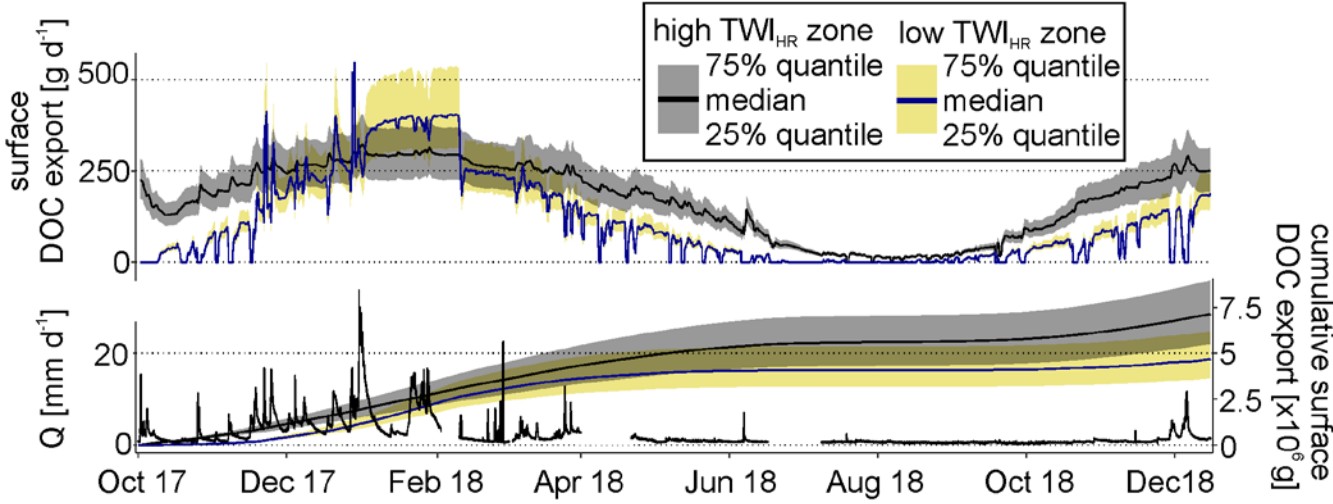

**Figure 7. Absolute DOC surface export (25, 50 and 75 percentiles) from high (DOC$_I$ quality) and low (DOC$_{II}$ quality) TWI$_{HR}$ source zones. Underlying DOC concentration percentiles based on all riparian groundwater samples of each source zone. Cumulative positive DOC export (25, 50 and 75 percentiles) from high and low TWI$_{HR}$ zones shown on the second y axis. Discharge rate (Q) (lower black line) shown on lower left vertical axis.**

## 4 Discussion

### 4.1 Small scale topographical heterogeneity delineates DOC source zones in terms of molecular composition and hydrologic connectivity

RZs of lower order streams constitute a manageable control unit for DOC export, but small scale processes in the RZ are not well enough understood to properly predict DOC export. We hypothesized that DOC production and transport are predominantly controlled by the micro-topography of the RZ (lateral variability), and by the location of the riparian groundwater level (temporal variability). Therefore we identified DOC source areas within a riparian zone based on topographic and chemical analyses and evaluated the temporally variable export of DOC from these source areas to the stream using numerical flow modeling.

In general, our chemical observations are in line with UV-Vis derived DOC properties presented in Werner et al. (2019) for the same study site. Other RZs in temperate (Strohmeier et al., 2013; Raeke et al., 2017), boreal (Ledesma et al., 2018b) and mediterranean (Guarch-Ribot and Butturini, 2016) climates show a comparable DOC property spectrum. Separating the riparian DOC samples according to their chemical characteristics revealed two chemically distinct DOC clusters in the RZ (DOC$_I$ and DOC$_{II}$). The molecular composition of DOC$_I$ indicates processed plant-derived organic matter, similar to typical wetland sites (Tfaily et al., 2018). The high DOC concentrations, oxygen-rich and aromatic character points to overall low organic matter turnover within the DOC$_I$ source zones. With that, DOC$_I$ is less available for microbial degradation (Mostovaya et al., 2017), but can readily be photodegraded (Wilske et al., 2020), and removed through sedimentation (Dadi et al., 2017) or in drinking water treatment (Raeke et al., 2017). In contrast, DOC$_{II}$ (more saturated, oxygen-depleted, and heteroatom

enriched) more represents microbial, secondary metabolites indicative of increased microbial processing of organic matter from organic rich top-soil layers and DOC which is not adsorbed to mineral phases. DOC of this chemical characteristics is typically found in deeper soil layers (Shen et al., 2015; Kaiser and Kalbitz, 2012; LaCroix et al., 2019). Yet, the DOC clusters were independent of sampling depth (note that a given sample location/ depth (Fig. S1) can experience low and high groundwater levels at different times). Rather, both DOC clusters were dividable by the $TWI_{HR}$ value of the well locations from which the respective samples were collected, suggesting the existence of two DOC pools with distinct source zones in the study site. Although the chosen threshold value is arbitrary and a small number of the wells switched from one DOC cluster to the other over time, separating DOC source zones based on $TWI_{HR}$ is plausible from a process-based perspective. High $TWI_{HR}$ values – connected to $DOC_I$ source zones – represent zones with increased water and solute (DOC) inputs from adjacent areas, leading to hydrological upwelling and DOC processing by micro-topography induced gradients in redox conditions (Frei et al., 2012). Zones with high $TWI_{HR}$ values are generally connected to higher mean groundwater levels due to increased flow convergence to these zones and concomitant waterlogging (Luke et al., 2007), which can create a conserving environment for DOC due to anaerobic conditions even in the topmost soil layers (LaCroix et al., 2019) potentially leading to DOC accumulation. In contrast, zones with low $TWI_{HR}$ values are generally characterized by lower mean groundwater levels and thus more oxygenated top soils, allowing for more biogeochemical processing. Overall we conclude that small-scale topographic heterogeneity (here represented by $TWI_{HR}$) can be linked to DOC properties in RZs, if the fluctuations of the groundwater level, which dynamically intersects with the surface micro-topography, are properly accounted for.

## 4.2 Connecting riparian DOC sources to the stream

The close agreement between molecular parameter of the $DOC_I$ and corresponding stream water samples in April and December suggest a predominant connectivity of $DOC_I$ source zones with the stream in times of high groundwater levels. In addition, the $DOC_I$ quality was similar between April and December indicating a DOC pool that is not strongly affected by seasonality and hydrologic conditions (i.e. steady export also during the wet and cold state, cf. Werner et al. 2019). Therefore $DOC_I$ can be regarded as a permanently available source of DOC to the stream water. In contrast, the $DOC_{II}$ molecular composition was reflected in the DOC composition of stream water in December but not in April, suggesting a connectivity of this pool during high flow periods but a potential depletion over time. Less organic matter input and lower biogeochemical process rates in winter and at the same time increased DOC export (at higher groundwater levels) may specifically deplete the $DOC_{II}$ pool (Werner et al., 2019). The molecular composition of $DOC_{II}$ samples in April thus may represent a pool of remaining DOC with low sorption affinity that was not depleted during high groundwater levels in winter (saturated and with larger molecular weight and oxygenation).

Variations in stream DOC molecular composition also appeared during events, indicating a link between DOC export and groundwater level dynamics which ultimately drive hydrological connectivity. Fully saturated riparian conditions connected the $DOC_I$ and $DOC_{II}$ source zones with the stream in December, leading to a stream DOC quality with molecular parameters

that were in-between the two riparian DOC clusters as also suggested by Werner et al. (2019) for this study site. On the other hand, $DOC_I$ and stream water DOC molecular compositions are converging during events with lower antecedent groundwater levels and unsaturated soils, suggesting that $DOC_I$ becomes the predominant source when $DOC_{II}$ is (still) depleted e.g. from snowmelt in April. We argue that changing load contributions from both riparian DOC source zones are induced by the prevailing hydrological situation, which distinctly shapes DOC export in this RZ. Observed deviations of instream DOC from

the riparian source DOC composition might be a result of near- and instream processing (Dawson et al., 2001; Battin et al., 2003), but also inter- and intra-annual variability of hydroclimatic drivers like seasonality or antecedent soil conditions (Werner et al., 2019; Köhler et al., 2009; Strohmeier et al., 2013; Futter and de Wit, 2008). Yet we showed direct links between six major DOC molecular properties and the DOC concentration of riparian and stream water samples. In comparison to other studies, which have used integrated or indirect signals to derive information on DOC characteristics, like (specific) UV

absorption and spectrophotometric slope values (Ledesma et al., 2018b; Werner et al., 2019), or electric conductivity and pH (Ploum et al., 2020), this allows a spatially explicit alignment of riparian DOC source zones to stream water samples at higher credibility.

Additionally, the physically-based model HGS independently showed that the dominant runoff generating mechanism (and thus potential DOC export pathway) at the study site is surface runoff from $DOC_I$ source zones. The high $TWI_{HR}$ zones are

characterized by morphological and hydrological conditions that produce more runoff per unit area and in turn they react more directly to precipitation than $DOC_{II}$ source zones (with low $TWI_{HR}$ values). The microtopography allows for re-infiltration of overland flow and exfiltration of shallow saturated flow. However, our model runs (and field observations) confirmed the importance of surface flow actually entering the stream as the dominant runoff generation mechanism. Even though surface flow was nearly absent during the summer, snowmelt in spring and abundant rainfall in the autumn both generated periods

with overland flow that were the main contributors to the annual runoff of the stream. Further confirmation that overland flow is the main delivery pathway of DOC to the stream can be found in the chemical fingerprints of individual source areas we sampled. Seven chemical characteristics of these sources (some of which are far from the stream) can be seen in the stream water as well, indicating that the hydrochemistry was not strongly affected when the water was carried from the source to the stream. This is consistent with a rapid delivery via overland flow, but not with a slower flow rate through the chemically

active subsurface. The volumetric range of event water in our catchment is in line with studies mentioned in Klaus and McDonnell (2013), but our results deviate from their general conclusion that stream water predominantly comes from pre-event water from the subsurface. Most of the runoff generation in our study site occurs during events under saturated conditions with significant fractions of fresh event water from direct precipitation onto saturated surface areas. In that sense the generated surface runoff is a mix of pre-event water that exfiltrates from the subsurface (as indicated in our study and

suggested by Frei et al. (2012)) with event water from precipitation. Surface contributions are linked to the landscape organization and topographic catchment characteristics (James and Roulet, 2009; Suecker et al., 2000). Our riparian study site has an overall low topographic relief (but steep hillslopes) and consists of hydromorphic soils of typically low hydraulic

conductivity. Given the nature of this study site, overland flow is to be expected as a significant export pathway connecting riparian DOC sources to the stream.

## 4.3 Quantifying riparian DOC export to stream

Quantifying DOC exports from riparian hot spots to the stream has rarely been done (Bernhardt et al., 2017), most probably because capturing mechanistic and spatio-temporal dynamics of small-scale hotspots and hot moments is challenging. Further, the impact of vertical versus horizontal heterogeneity on solute export is still under debate, although mostly at the large scale (Herndon et al., 2015; Zhi et al., 2019). We found no statistical significant relation between sampling depth and well DOC classification in our samples, but the possibility of a bias exists since samples were predominantly taken in deeper soil layers – also due to the fact that there often was no surface-near water available when groundwater samples were taken. Yet, surface-near samples as well as deep samples appear in both clusters at more or less equal parts. The presence of micro-topography induced hot spots in RZs has been theoretically shown in a modeling study by Frei et al. (2012), who described a small-scale hydraulic and biogeochemical impact on nutrient processing in a generic wetland. In this study, we showed an overall dominance of $DOC_I$ export from high $TWI_{HR}$ zones during events, despite making up only about 15 % of the total study site area. Zones with a high $TWI_{HR}$ export 8.7 times more DOC than low $TWI_{HR}$ zones, which is in line with DOC export from hotspots in a forested stream in a humid continental climate (Wilson et al., 2013) and generally in range of different hot spot effect sizes (Bernhardt et al., 2017). Based on our spatio-chemical classification and numerical water flow modeling, our findings suggest a clear link between DOC quality and lateral topographic heterogeneity (represented by $TWI_{HR}$) supporting the findings from Frei et al. (2012). As with Ledesma et al. (2018b) our work recommends to focus riparian DOC research on overall smaller scales, but in addition we showed that surface flow from localized source areas is an important carrier of DOC towards the stream. The results obtained from our study site therefore provide no direct evidence for the existence of a distinct dominant source layer in the subsurface (Ledesma et al., 2018b) and put a spatially lumped application of a riparian integration model as proposed by Seibert et al. (2009) into question. Both these other approaches assume predominantly horizontal subsurface flow as the main transport mechanism for DOC. At our site in contrast the shallow depth to bedrock and the humid climate facilitated the creation of generally wet conditions, where saturation-excess overland flow can become the main mode of DOC export to the stream. Micro-topography determines the location (lateral variability) of the dominant DOC source areas, but the dynamic mobilization of DOC during events is controlled by the hydrological drivers rainfall and snowmelt and their immediate effect on riparian groundwater levels (temporal heterogeneity).

## 4.4 Potential for future work and implications

Recent studies have concluded that lateral DOC fluxes from terrestrial to aquatic ecosystems are not well researched, but contribute an important share of the global DOC budget (Zarnetske et al., 2018; Wen et al., 2020). In this regard, we found that surficial DOC export dominated overall DOC export to the stream at our study site. Yet surface DOC export is underrepresented in current model-conceptualizations of lateral DOC export (Dick et al., 2015; Ledesma et al., 2018a; Bracken

et al., 2013). Reasons for the exclusion might be due to the high complexity of representing the spatio-temporal heterogeneity of surface export in modeling concepts or just because it may play a minor role in other catchments. Considering highly resolved heterogeneity in large-scale models is not tractable, but understanding the export mechanisms of riparian zones at high spatial resolution allows to better estimate overall DOC export potential of catchments as a function of climatic variability and general topographic structure (Jencso et al., 2009). Further investigating $TWI_{HR}$ dynamics could lead to new approaches

and concepts for better DOC export modeling: A general, coarse-scale relationship between soil moisture and potential surface runoff generation has already been proposed based on a catchment-scale topography-driven runoff proxy (Gao et al., 2019, Birkel et al., 2020). The mechanistic connection between $TWI_{HR}$ and surface DOC export in our study represents a similar general mechanism. This mechanism is potentially applicable to the entire riparian zone in small catchments (similar $TWI_{HR}$ values should result in similar runoff generation), where stream runoff generation and DOC mobilization are inherently

coupled. Respective model fluxes could be validated by each other thereby ensuring correct internal model functioning. Implementing the described DOC surface export mechanism to coupled hydrological-biogeochemical DOC export models (Dick et al., 2015; Lessels et al., 2015) could greatly improve the mechanistic basis of threshold based and non-linear model fluxes (Birkel et al., 2017). Combining a good mechanistical biogeochemical and hydrological model representation with knowledge of DOC properties in respective DOC source zones could further reduce the need for tracer experiments in the

field. The simultaneous simulation of fluxes of water and DOC of such a model combination facilitates internal model calibration and reduces equifinality (Abbott et al., 2016; Birkel et al., 2020), potentially leading to a more accurate upscaling of DOC export from RZs, especially for those with mild slopes and high groundwater levels.

## 5 Summary and Conclusions

To elucidate spatial and temporal variability of DOC exports from riparian zones this study combined a hydro-morphological
classification of a humid temperate riparian zone with a chemical characterization of DOC in source zones and the stream and detailed modelling of water fluxes and flow paths. The chemical classification of riparian water samples via ultra-high resolution FT-ICR-MS revealed two distinct DOC clusters ($DOC_I$ and $DOC_{II}$) in the riparian zone. Degrading plant material presumably contributes most to an aromatic, oxygen-rich DOC with high concentrations ($DOC_I$ cluster), located in regions of high wetness in local topographic depressions. The $DOC_I$ is available for photo-degradation, and can be relatively easily

removed through sedimentation or later during drinking water treatment. The second cluster ($DOC_{II}$) reflects microbially processed, nonpolar (yet mobile) DOC with lower concentration and larger compositional variability across seasons that may be more persistent in the surface water. The source zones of $DOC_I$ and $DOC_{II}$ within the riparian zone can be separated and mapped by setting a threshold value for the high-resolution TWI ($TWI_{HR}$), suggesting the existence of two distinct DOC pools. The identification of source zones was achieved via independent measures (unsupervised chemical classification, and TWI-

based physical flux modeling) leading to higher credibility. The hydrological modeling revealed that the dominant runoff generation mechanism at the study site was surface runoff. We were able to quantify the contribution of each source zone to

DOC export and noticed significant variations in those depending on the groundwater level and seasonal variations in the degree of depletion of the $DOC_{II}$ pool. Mostly during events, $DOC_I$ source zones, which make up only 15 % of the riparian area exported 1.5 times more DOC than the remaining 85 % of the area associated with the $DOC_{II}$ source zones. Highest discrepancy between the loads exported from the $DOC_I$ and $DOC_{II}$ zones was evident for events at intermediate antecedent wetness states (neither completely saturated nor very dry conditions). Overall we can confirm our initial hypothesis that DOC production and export are controlled by the lateral variability of topography within the RZ interplaying with the temporal variability of groundwater levels. Understanding the export mechanisms of riparian zones at high spatial resolution allows to better estimate overall DOC export potential of catchments as a function of climatic variability and general topographic structure. In contrast to other studies in boreal catchments, this study highlights that DOC export from the riparian zone by surface runoff is important for DOC export from hydromorphic soils with overall low topographic riparian relief. Surface export should therefore not be neglected in DOC export models. Delineating activated source zones for DOC export by suitable proxies of the micro-topography (here represented by $TWI_{HR}$) can help to identify source zones in existing DOC models as well as surface flow pathways from these sources to the stream. The combined biogeochemical and hydrological understanding of riparian DOC export generated here can be implemented in models. By doing so, fewer field tracer experiments may be needed while model calibration improves. This makes it easier to apply such models to a variety of RZs (especially those with mild slopes and high groundwater levels), possibly leading to catchment-scale model-based estimates of DOC-export. This work showed that micro-topography and the groundwater level are important determinants for DOC production and transport in a RZ. The micro-topography varies only slowly with time, and can be captured in a DEM (e.g determined by a suitably equipped drone). Such a DEM enables to identify probable source areas of DOC as well as potential flow paths that connect the source areas to the stream. Disrupting either the source areas or the flow paths, or both, can potentially reduce the DOC loading of the stream at low cost and with relatively little disturbance of the local ecosystem. Such measures might have to be repeated every few years, but even then, they may provide an attractive low-cost method to manage DOC in streams.

*Data availability.* All datasets used in this synthesis are publicly available via the following link: https://doi.org/10.4211/hs.b32ba184414e475ba36a0bb193866ef1.

*Supplement.* The supplement related to this article is available online at:

*Author contributions.* BJW, JHF, OJL, AM and GHdR planned and designed the research. BJW performed the statistical analysis and wrote the paper with contributions from all co-authors. JY performed the hydrological modelling. BJW and OJL analysed and interpreted the FT-ICR-MS data. UW implemented geophysical investigations. RG provided post-processed drone altimetry data.

*Competing interests.* Gerrit de Rooij is a member of the editorial board of the journal.

*Acknowledgements.* Jan M. Kaesler is gratefully acknowledged for the FT-ICR-MS analysis at the Centre for Chemical Microscopy (ProVIS) at the Helmholtz Centre for Environmental Research, which is supported by the European Regional Development Funds (EFRE-Europe funds Saxony) and the Helmholtz Association. We thank Toralf Keller for excellent and steady field work, Marco Pohle for geophysical field work, Kai Franze for software development and Heidrun Paschke and Michaela Wunderlich for help with the DOC and nutrient analysis.

*Financial support.* This research was funded by the Helmholtz Research Program POF-III, Integrated Project "Water and Matter Flux Dynamics in Catchments".

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
