# Peer review of "Small-scale topography explains patterns and dynamics of dissolved organic carbon exports from the riparian zone of a temperate, forested catchment"

_Hydrology and Earth System Sciences, 2021_

## Author Comment (AC1)

This study is an impressive assemblage of field, laboratory and modelling techniques to determine the spatio-temporal variability in DOC export (concentration and molecular composition) in a riparian zone. The abundance of different techniques make the manuscript quite dense and it is sometimes difficult to follow the details of the Material and methods, but I would not recommend providing more technicalities (see some exceptions in the detailed suggestions below).

My main suggestion to improve the paper is to rework the introduction and the discussion to 1) identify a clear research question or hypothesis, the introduction is lacking a "problem to solve". It was not clear to me why such a detailed study would improve management, because the resolution is far higher than any management action, or large scale modelling; 2) develop a discussion that put the results into perspective, while the current discussion is still very similar to the result section and does not contain implications for future research or management.

We appreciate your constructive, detailed evaluation of our Manuscript (MS). We realized that our research question/hypothesis was not formulated clearly enough, which is also reflected in the reviewers statement that there is a lack of a "problem to solve" in the introduction. This will be changed and addressed in the introduction and discussion sections of the MS. In line with the proposal of Referee #1 (comment on L80), we will outline our main hypothesis, that a small-scale, dynamic TWI can explain the main characteristics of the DOC export regime of a given RZ, more precisely. This hypothesis will be tested by a combination of field measurements and detailed hydrological modelling. We will furthermore carefully rework the introduction, discussion and conclusion with regard to implications for management and will put our research into a broader perspective of literature (e.g. references given in the review of Referee #2). As this rewriting needs to be done carefully and requires some work we are not yet able to propose specific changes at this time.

I also found that the authors made too little use of the different sampling dates, especially those during storm events. I did not understand why several analyses in the manuscript only consider April and December dates, while many other dates are available (I may have missed something here…). Similarly, a high-temporal resolution sampling was performed during selected storm event but the infra-storm events dynamics is not described.

Regarding the riparian water sampling, only 2 more samples are available for July (cf. Figure 3b). Generally groundwater sampling in summer turned out to be difficult due to low groundwater levels. Most wells, especially those screened closer to the surface, could not be sampled in summer. To ensure proper comparability, we decided to focus on April and December, when groundwater and surface water sampling was possible. Regarding the high resolution event sampling, all samples were used (cf. Figure 4a), but inter-event variance of DOC properties is higher than intra-event variance. Therefore we considered bulk sample properties of one event to be satisfactory in information content. We will state this more explicitly in the MS.

Some work would be necessary to improve the clarity of the text: shorter sentence, less and better use of conjunctions, correct some poor phrasing. I have identified a few examples of sentences to improve but note that English is not my native language either.

We will work through the MS to identify sentences similar to those identified by the reviewer and change them accordingly.

Detailed comments:

Title : "from a riparian zone of a" -> "from the riparian zone of a"?

We agree, the title will be changed accordingly.

L12 "but poorly understood component": what specifically is not understood. Identify a "problem to solve", a research question or hypothesis in the abstract.

We agree, the sentence will be changed to:

The mechanisms of dissolved organic carbon (DOC) export from riparian zones (RZs) is an important, yet still poorly understood component of the carbon budget in catchments with a temperate climate.

A "problem to solve", a research question or hypothesis will be more concisely stated in the abstract. In line with the proposal of Referee #1 (comment on L80), our main hypothesis is that a small-scale, dynamic TWI can explain the main characteristics of the DOC export regime of a given RZ. This hypothesis is tested by a combination of field measurements and detailed hydrological modelling.

L15 "high spatio temporal resolution": what is the resolution of the DEM?

This information is given in the following sentence (1m). Therefore this sentence will not be adapted.

L15 "Stream water DOC samples from differing hydrological situations": describe these situations, number of sampling dates, study period, etc in the abstract.

We agree, the sentence will be changed to

From May 2017 until July 2019, stream water DOC samples (n = 73; five runoff situations and five grab samples) were compared to riparian DOC groundwater and surface water samples (n = 66) …

L18 "were then simulated": avoid passive voice throughout the manuscript

We agree, the sentence will be changed to

We used a physically-based, fully-integrated numerical flow model (HydroGeoSphere) to simulate explicit water fluxes from the resulting riparian DOC source.

L20 "two distinct DOC pools (DOCI and DOCII)": describe what make them different in the abstract

We agree, the sentence will be changed to

Chemical classification revealed an aromatic, oxygen-rich DOC pool with high concentrations ($DOC_I$) and a microbially processed, mobile DOC pool with lower concentrations but larger compositional variability across seasons ($DOC_{II}$) in the RZ.

L22 "high-resolution topographical wetness index (TWIHR)": specify resolution in abstract.

We agree, the resolution will be specified accordingly.

L27 "should be considered in DOC export models": any implications for management? Should large-scale models really consider this fine-resolution heterogeneity?

Considering fine-resolution heterogeneity is likely not applicable to large-scale models, but understanding the export mechanisms of riparian zones at fine scale allows to better estimate overall DOC export potential of catchments as a function of climatic variability and general topographic structure. However, the relationship between riparian zone structural heterogeneity and DOC export presented in this study suggests that knowledge about riparian topography and structure, easily derived from DEMs, may be useful for the development of more parsimonious models for the prediction of hydrologic and DOC export response by e.g. implementing a threshold-based surface runoff module. Measures of riparian zone source connectivity (like the presented TWIHR) provide an integrated measure of riparian zone surface runoff generation and the associated DOC export behavior that is – when integrated accordingly – scalable to catchment level. We therefore believe that proxies of fine-resolution structural heterogeneity can improve large-scale catchment models, which cannot represent topographic (riparian) relief at very fine scale. We agree with the referee that our detailed, fine-resolution modelling effort will generally not be feasible for day-to-day management operations. However, it helped us to demonstrate the usefulness of drone-based DEMs to gain process understanding. With the help of the model we could illustrate the importance of the TWIHR for DOC export. A practical consequence of our work could be that the availability of a high-resolution DEM allows one-time targeted earth-moving measures to manipulate the TWIHR to create a more favorable DOC export regime.

We will address this in the Introduction and Discussion of the MS and thus will modify this sentence.

L27 "But despite": don't start a sentence with "but"

We agree, please see next comment.

L33 "but could" second but in this long sentence

We agree, the sentence will be changed to

Changes in land use, climate and biogeochemical boundary conditions have increased DOC concentrations in surface waters and changed the quality of the exported DOC (Larsen et al., 2011; Chantigny, 2003; Wilson and Xenopoulos, 2008). Routine management of DOC could help to comply with water quality directives and lower the cost of drinking water purification (Matilainen et al., 2011), but is currently almost non-existent (Stanley et al., 2012).

L36 "Especially riparian zones (RZs) of lower order streams are potential targets for…" poor phrasing

We agree, the sentence will be changed to

Lower order streams make up a large fraction of the total river networks worldwide (Raymond et al., 2013) and their riparian zones (RZs) represent a main source for terrestrial DOC export (Ledesma et al., 2015; Musolff et al., 2018). Therefore RZs of lower order streams – as terrestrial-aquatic interfaces – constitute a general control unit qualifying them as potential targets for DOC export management.

L41 "Here, DOC …" add reference

We agree, the following reference will be added:

Luke, S. H., Luckai, N. J., Burke, J. M., and Prepas, E. E.: Riparian areas in the Canadian boreal forest and linkages with water quality in streams, Environmental Reviews, 15, 79-97, 10.1139/A07-001, 2007.

L46 "This leads to a stronger accumulation of DOC close to the soil surface…" I did not understand the link with the previous sentence

We agree, the sentence will be changed to

On the other hand, the rate of DOC accumulation and ultimately export is also dependent on hydrological connectivity of DOC sources to the stream. Between events, water cannot mobilize existing DOC pools close to the soil surface. Consequently, hydrological connectivity also contributes to DOC accumulation close to the soil surface in comparison to deeper soil layers that are more frequently connected with the stream.

L49 "led to concepts like variable source zone activation (Dick et al., 2015; Werner et al., 2019), the dominant source layer (Ledesma et al., 2015) and transmissivity feedback (Bishop et al., 2004)" explain these concepts and their limits. Listing them is not enough in an introduction

We agree, the concepts will be explained accordingly. The sentences will be changed to

Several attempts have been made to acknowledge the vertical variability of lateral DOC transport to streams. The dominant source layer concept (Ledesma et al., 2015) focuses on depth-dependent differences in DOC pools in different soil layers. Transmissivity feedback (Bishop et al., 2004) accounts for depth-dependent differences in hydraulic conductivities of soils and the resulting changes in the transmissivity of the soil profile under changing groundwater levels. Variations in the lateral hydrological connectivity of RZs were coneptualized by variable source zone activation (Dick et al., 2015). These concepts still describe a heterogeneous system in terms of an integrated response without acknowledging the distinct spatio-temporal variability in DOC export of single landscape units (Ledesma et al., 2018a; Ploum et al., 2020; Dick et al., 2015).

L55 "a strong focus on vertical heterogeneity" I my understanding the variable source area concept is more about horizontal heterogeneity.

We agree, the text will be changed to (please also see comment above)

Variations in lateral hydrological connectivity of RZs were conceptualized by variable source zone activation (Dick et al., 2015).

L57 "Moreover RZs are highly dynamic and heterogeneous with micro-topography" the role of micro topography is central to the hypothesis of this work and should be better highlighted.

We agree, the role of micro-topography will be better highlighted. We propose to change the text to

Moreover micro-topography in RZs can induce hot spots of biogeochemicall activity (Frei et al., 2012) that contribute disproportionally strong to nutrient turnover. Furthermore, micro-topography focuses drainage (Frei et al. 2010, Scheliga et al., 2019) and consequently solute export to the stream when nutrient hot spots get hydrologically connected to the stream (during hot moments). Therefore, micro-topography in the RZ is considered a fundamental organizing structure of soil chemistry (Diamond et al., 2019) and hydrological connectivity (Scheliga et al., 2019) that induces high spatio-temporal heterogeneity.

Additional references:

Diamond, J. S., McLaughlin, D. L., Slesak, R. A., and Stovall, A.: Microtopography is a fundamental organizing structure of vegetation and soil chemistry in black ash wetlands, Biogeosciences, 17, 901-915, 10.5194/bg-17-901-2020, 2020.

Scheliga, B., Tetzlaff, D., Nuetzmann, G., and Soulsby, C.: Assessing runoff generation in riparian wetlands: monitoring groundwater–surface water dynamics at the micro-catchment scale, Environmental Monitoring and Assessment, 191, 116, 10.1007/s10661-019-7237-2, 2019.

L65 "Model conceptualizations that are able to bridge those scales" with this sentence it seems that the paper will deal with this question of scales, but it is not the case.

We agree, the sentence will be removed from the manuscript.

L80 "We argue that a smaller-scale, dynamic assessment of the TWI…" should be the hypothesis of the paper. Please give a response to his hypothesis/question in the discussion/conclusion.

We agree, a dynamic assessment of the high-resolution TWI will be added in the regarding discussion/conclusion section. We therefore hypothesize that a small-scale, dynamic TWI can explain the main characteristics of the DOC export regime of a given RZ. This hypothesis will be tested by a combination of field measurements and detailed hydrological modelling.

L97 "In this paper we…" intro long enough, no need for a summary of the methods here. Develop problem to solve instead.

We agree, the methodological shares of this paragraph will be deleted and instead replaced by a problem to solve (see comment above).

L100 "More specifically, (1)…" it would be better to list specific research questions than summary of the methods.

We agree, we will replace the summary of the methods by specific research questions/hypotheses.

L128 "Electric resistivity tomography (Resecs DC resistivity meter system, Kiel, Germany) was applied at two transects" show the transects in figure 1?

We agree, transects of the conducted electric resistivity tomography will be shown in figure 1b. The captions will be adapted accordingly.

L134 "Two PCM4 portable flow meters (Nivus, Germany) measured discharge in the Rappbode stream at a chosen inlet…" show inlet and outlet in figure 1?

We agree, locations of in- and outlet will be shown in figure 1b.

L151 "To have maximum ability in capturing the magnitude and direction of this slope…" poor phrasing

We agree, the sentence will be rephrased to

Therefore we decided to install a piezometer network aligned on a square grid, with one principal axis oriented in parallel to the stream and the other perpendicular to the stream to capture the temporal dynamics of the groundwater level in both principal directions of this slope.

L156 "In addition 3 more wells were installed at 0.3 m depth inside the rectangular grid for surface near sampling." I did not understand this sentence.

We agree, we will change the sentence to

We installed three additional wells with screens at 0.3 m depth inside the rectangular grid for sampling near the surface.

2.2.3. I found it difficult understand the maximum depth and the screening height of the different piezometers and wells. Please rework this section to improve clarity.

We agree, the section will be reworked with focus on improved clarity.

L166 "Biweekly routine samples…" it is never clear whether biweekly means twice a week or every second week. Please use a less ambiguous term. Please also add the number of sampling dates and the number of dates when FT-ICR mass spectrometry was used. Is it only two dates?

We agree, the number of sampling dates will be added. We further realized that the referee was confused about the description of FTICRMS samples. We therefore decided to additionally rephrase the whole paragraph as follows:

At the monitoring site along the Rappbode stream, we collected 68 stream event samples during five events, 66 riparian samples and 5 stream samples during 5 occasions (one stream sample per occasion), which were analyzed for DOC concentration. The molecular composition of the DOC was determined via FT-ICR MS (see Table S1 for detailed information). Auto-samplers (6712 Full-Size Portable Sampler, Teledyne ISCO, US) were triggered by the rate of water level increase to sample stream water during discharge-generating events at least once per hour. Auto-sampler bottles (PP) were soaked for 48 h in 0.1 N HCl prior to use. We prepared process blanks with deionized water to correct for eventual contamination during field work and sample processing. Due to the remoteness of the study site, we collected auto-sampled stream water samples within 4 days after the triggered event sampling. Samples were stored in the dark inside the sampler and air temperatures were below 10°C during that time. We collected riparian zone shallow groundwater samples from 3 to 18 out of the 28 installed piezometers depending on hydrological conditions during the five sampling dates. Before sample collection, we replaced water in the wells one to three times (based on the responsivity of the wells) through pumping. We rinsed the flasks and the pump with sample water prior to sample collection and subsequently transferred 100 mL sample into acid-rinsed (0.1 N HCl) and baked (500 °C, 4 h) glass bottles. Additional stream water was collected for each riparian sampling date. Moreover, we collected 38 routine samples in the Rappbode stream (every two weeks) and processed them accordingly to determine DOC concentration. Samples were stored dark and cool until further processing in the laboratory.

L181 "samples were filtered using 0.45 μm membrane filters" did you filter the samples in the field or back in the lab?

We agree, the sentence will be changed to

Samples were filtered (0.45 µm membrane cellulose acetate filters, Th. Geyer, Germany) and acidified to pH 2 (HCl, 30 %, Merk, Germany) on site. Subsequently samples were stored cool until DOC measurement and extraction in the laboratory.

L280 "2.4.3 Calibration" is it possible to provide the objective function of the calibration? I understood that the model aimed to simulate both the stream discharge and groundwater depth in several wells, with a weighting scheme giving a high importance to the groundwater, but it would be interesting to see the equation of this objective function.

We agree, the objective function will be provided in the SI:

$$\text{Multi} - \text{Objective function} = \sum_{i=1}^{i=nq} w_q (O_q^i - S_q^i)^2 + \sum_{i=1}^{i=nl} w_l (O_l^i - S_l^i)^2$$

Where $O_q^i$ and $S_q^i$ are the observed and simulated discharge. $nq$ is the number of number of the discharge observation (611). $O_l^i$ and $S_l^i$ are the observed and simulated groundwater level. $nl$ is the number of groundwater level observation (110140). $w_q$ and $w_l$ are the weights for the two observation groups, both being assigned with the value of 1 in the calibration. Because the observation number of groundwater level was significantly larger than that of the discharge, this multi-objective function highlight the importance of the groundwater levels, such that the 94% of the multi-objective function for the calibrated best-fit was attributed to groundwater levels.

L322 "The DInf algorithm was used" please explain what it is.

We agree, the DInf algorithm will be explained.

We applied the DInf algorithm to calculate a realistic hydrological routing (Tarboton, 1997). The DInf algorithm determines flow direction as the steepest downward slope on eight triangular facets formed in a 3x3 cell window centered on the cell of interest.

L335 "Discharge shows event-type, erratic variability" poor phrasing

We agree, the sentence will be changed to

Discharge showed typical erratic variability at the event-scale. At annual scale, discharge expressed a clear seasonal pattern, with lowest values in late summer and highest values in spring (Figure 3a).

L360 "DOC in riparian water samples was in general of highly unsaturated and phenolic composition, typically found in lignin and biomass type compounds" can we see this in a table or a figure?

This can be derived from Figure S10. We have noticed that the colors in this Figure were wrong and will include a corrected version of Figure S10:

[Figure]

Fig. S10: Aggregated van Krevelen plot of all FT-ICR-MS sample of stream (red) and riparian (black) origin. Data represent the intensity weighted average of the molecular H/C and O/C ratios considering all valid MF in these samples.

Besides, the sentence will be changed to

DOC in riparian water samples was in general of unsaturated and phenolic composition ($wa_{HC}$ = 1.27 ± 0.05; $wa_{OC}$ = 0.40 ± 0.01; n = 66), typically found in wetland surface soils (LaCroix 2019). However, stream event samples significantly differed (p < 0.001) from riparian samples and were more unsaturated ($wa_{HC}$ = 1.17 ± 0.05; n = 76) and more oxygenated ($wa_{OC}$ = 0.43 ± 0.03) as shown in Figure S10.

L395 "Note that wells, sampled during different occasions throughout the year occur in both DOC clusters and according TWIHR values can thus occur in both clusters" it is unclear to me to what extend a given piezometer belonged to the same cluster throughout time. This sentence suggests that the cluster can change, but a quantitative assessment of how many piezometer remain in the same cluster or change clusters would be interesting here.

We agree, a quantitative assessment of how many piezometer remain in the same cluster or change clusters will be added.

L415 "The significant difference in TWIHR median values of DOCI and DOCII wells" I did not understand how you could classify wells as DOCI-well or DOCII-well if a given well could change clusters in different dates.

We agree, the correct term in this sentence is DOCI and DOC II *samples*, not *wells*. We will change this in the manuscript and apologize for the inconvenience.

L416 "using the median TWIHR value of the DOCI group (9.66) as a threshold." I did not understand this choice; please explain the rationale behind this.

The rationale was to map DOC source zones of different DOC concentration and chemical properties. We found an overlap of TWIHR values between both groups although their median was found to be significantly different. Using the median of DOCI (9.66) as a manually chosen threshold we can separate both groups capturing 50% of all cases of group I in one class while only allowing 25% of group II.

We realized that the rationale behind the selection of the threshold was not addressed clearly enough in the MS. Therefore we will better describe and clarify the rationale.

L418 "Also note that different samples of one well can appear in both DOC groups" please give numbers.

We agree, numbers will be added in the manuscript.

L435 "Fig. S7, Table S4 for according water fluxes" -> "corresponding water fluxes"?

We agree, this will be changed in the manuscript.

L454 "During the model period, DOCI source wells had a median DOC concentration of 5.8 mg L-1 which was 2.3 times higher than for the DOCII source wells" it would be interesting to remind the mean+/- sd of the two types of wells. Do deeper wells match with the DOCI cluster?

We agree, $\pm$ sd will be added:

During the model period, DOCI source wells had a median DOC concentration of 5.8 mg L-1 (mean $\pm$ sd: $6.2 \pm 2.7$ mg $L^{-1}$) which was 2.3 times higher than the median for the DOCII source wells ($2.7 \pm 1.2$ mg $L^{-1}$).

Certain deeper wells match with the DOCI cluster (e.g. A1-E1, B4, C4, see also comment below).

L487 "as typically found in deeper soil layers" what influences the difference between DOC I and DOC II more: the TWI or the sampling depth? (or both are related?).

There is no statistical significant relation between sampling depth and well classification in our samples (see Figure below). On the other hand, we presented significant differences in TWIHR between the clusters. We therefore conclude that TWI (as postulated in the MS) controls the difference between DOCI and DOC II (more).

However, the possibility of a bias exists since samples were predominantly taken in deeper soil layers - also due to the fact that there often was no surface near water available when

groundwater samples were taken. Moreover, surface near samples as well as deep samples appear in both clusters.

Also with regard to referee #2 we will include this discussion and present respective statistical test results in the MS.

[Figure]

Figure: Boxplots of piezometer slot/sampling depth [cm below ground] and TWI [-] value for the DOC$_I$ and DOC$_{II}$ clusters. Horizontal brackets above describe the Kolmogorov-Smirnoff (KS) and Wilcoxon rank sum (WC) test statistics. Values were min-max normalized to values between 0 and 1 for better illustration.

L491 "indicating a replete DOC pool with constant contribution to the overall DOC quality in the stream" unclear sentence

We agree, the sentence will be changed to

In addition, the DOCI quality was similar between April and December indicating a large DOC pool which is less affected by seasonality and hydrologic conditions. Therefore DOCI can be regarded as constantly contributing to the overall DOC quality in the stream.

L493 "indicating the influence of seasonality on this pool." It is difficult to make such a conclusion with only two dates.

We agree, the sentence will be changed to

In contrast, the DOCII composition was reflected in the stream water composition in December but not in April, suggesting a depletion of this pool (e.g. induced by seasonality) during high flow periods.

General comment on "4 Discussion": this discussion is too similar to the result section, many conclusions are specific to the study site while readers would expect to see the results put into perspective, with more implications for management and research, more key messages and more references to the literature.

We agree, the Discussion section will be changed accordingly. In consistence with the Introduction (see general comment and comments on L12, L27), results will be put into a broader perspective in reference to other studies reported in the literature. Implications for management and research will be given more weight, key messages (see also comment L80) will be better worked out. The suggestions provided in the detailed comments by both referees will offer useful guidance for this. Because this rewriting needs to be done carefully and requires some work we are not yet able to propose specific details of these changes at this time.

---

## Author Comment (AC2)

Werner et al. examines export patterns and dynamics of dissolved organic carbon from the riparian zone in a temperate, forested catchment. The paper used an array of different approaches to relate DOC source zones within the RZ to their dominant DOC export mechanisms. Stream DOC samples from different hydrological conditions were compared to riparian DOC groundwater and surface water chemistry. They also characterized DOC chemically (via Fourier-transform ion cyclotron resonance mass spectrometry) and used topographic analysis (at a resolution of 1m). Water fluxes were simulated using the code HydroGeoSphere. The paper concluded that surface runoff from zones of high TWIHR values, which occupied about 15% of the total area, exported about 1.5 times the load of DOC from the remaining 85 % of the area, and that "this study highlights that surface DOC export from the riparian zone plays an important role for lateral DOC export from hydromorphic soils with overall low topographic relief."

The work is interesting and collated an array of approaches from chemical analysis to modeling at high spatial resolution. The current manuscript is phrased from the angle of horizontal heterogeneity / landscape topography, which I think is actually already well studied [Herndon et al., 2015; Jencso et al., 2009; Ledesma et al., 2018; McGuire and McDonnell, 2010; Pacific et al., 2010]. But I find the conclusion is not particularly surprising.

We appreciate your evaluation of our Manuscript (MS). Your comments as well as those by the other reviewer make it clear that we need to phrase our conclusions more crisply and clarify the contribution of our work to the existing body of knowledge and its practical implications (see also Referee #1, General Comment on "4 Discussion"). We will discuss the references you mention if we are allowed to revise our paper.

Please note that most of the references you mention (Herndon et al., 2015; McGuire and McDonnell, 2010; Klaus and McDonnell, 2013; Zhi and Li, 2020; Zhi et al., 2019) are concerned with much larger scales than our work, which studies an individual riparian zone (RZ) in detail. The scale of our experimental site is the smallest on which a hydrological landscape unit (in this case, the RZ) can be studied in its entirety and the largest that still allows extensive monitoring in the field during several seasons within realistic constraints on resources and personnel. The paper's contribution lies in the combination of multi-sensor field monitoring backed up by detailed hydrological modelling and high-resolution chemical analyses of DOC quantity and quality. This allowed us to determine the size and contribution of hydrologically different DOC source areas, something that so far has only rarely been quantified (Bernhardt et al., 2017). We intend to better clarify this in the text and improve the explanation of the practical relevance of some of our findings. Perhaps an improved phrasing of our results and the conclusions we draw from them can illuminate where we agree or disagree with earlier work. For instance, Ledesma et al. (2018) contradict the references above by stating that the RZ, not the hillslope, is the dominating source of DOC in streams. Our work supports that claim but shows that surface flow is an important carrier of DOC stemming from localized source areas, in contrast with the concepts of a dominant source layer in the subsurface (Ledesma et al., 2018) or a riparian integration model (Seibert et al., 2009) that both hypothesize predominantly horizontal subsurface flow as the main transport mechanism for DOC. This suggests – as also assumed by Jensco et al., 2009 – a dependency of DOC export on morphologic, climatic and topographic conditions. Our monitoring set-up is well suited for identifying such conditions.

On the other hand, it seems to me that this work presents a rare opportunity to dig deeper to think about the relative influence of vertical versus horizontal heterogeneity. The relative importance of vertical versus horizontal heterogeneity in doc export is poorly understood. In particular, there has been quite some interests in understanding the solute export from different subsurface depths, for example, [Seibert et al., 2009; Zhi and Li, 2020; Zhi et al., 2019]

The relative influence of vertical versus horizontal heterogeneity will be discussed in our MS (see also Referee #1, L487). As mentioned above, we found that surficial DOC export from high TWI zones dominates DOC export from riparian zones to the stream at our study site suggesting that horizontal heterogeneity predominates vertical heterogeneity in low relief catchments with hydromorphic soils. We also point out that the top few decimeters of soil was very organic and contained many rhizomes. Somewhat deeper, the soil contained so many rock fragments that digging with a spade was no longer possible. At roughly 1 m (with considerable variations), fractured bedrock occurred. To study solute export from various depths in the field, tracer experiments are needed. However, taking samples at well-defined locations in this soil proved very difficult, even in the dry season when the groundwater was not near the soil surface. We do not believe it is feasible to excavate the soil in intervals of 5 or 10 cm to study the distribution of a dye tracer. It is likely that colleagues elsewhere have experienced similar difficulties. This would explain why the relative importance of vertical versus horizontal heterogeneity in doc export is poorly understood. We fear that may remain so for the foreseeable future.

The data from this work have depth profile (top 100 cm) of doc, and flow calcination from different depths. These two can be combined to calculate at what depth most doc was exported, and how the export varied with depth in high flow events. At a minimum, it would be nice to see some discussion along this line of vertical heterogeneity.

We agree, some discussion along this line will be added to the MS (see also comments below and specific response to comment of referee 1, L487). However, given the difficulties associated with tracer experiments to identify flow paths in this soil that we outlined above, this discussion will have to be based entirely on the numerical model and therefore has to rely on a schematized conceptualization of the subsurface.

I also find "Surface export" is a confusing term. Is this really surface runoff, or does the water mostly flow through top soil? Unless in extremely large events, most forests do not see significant amount of surface runoff. In many places, stream water comes from "old" water from the subsurface, not surface runoff "new" water [Klaus and McDonnell, 2013].

There is no ambiguity in our use of the term. The RZ on which this paper focuses was not forested, although the surrounding slopes were. Overland flow in the RZ was quite common during wet periods. With "surface export" we refer to water that has been on the surface at least once on its way to the stream – as indicated by respective exchange fluxes. We will clarify this in the text. Surface flow can reinfiltrate and flow to the stream/ boundaries through the subsurface.
The steep hillslope but low slope and hydromorphic soil in the study site; micro-topography as well as scale and climatic setting of our field site contrasts those reviewed by Klaus and McDonnell (2013). Also, they view water contributions from a watershed/catchment-scale perspective, whereas we focus on the RZ itself. Given the nature of our study site, more overland flow is to be expected (although Klaus and McDonell also reviewed studies that had

up to 100% "new" water contributions). We explicitly mentioned that our findings hold for a low relief riparian zone with hydromorphic soils to account for these differences.
As discussed below (comment on L525 and following) we therefore do not think that the concept presented in Klaus and McDonnell holds for our field site.

Line 52-54, "a strong focus on vertical heterogeneity". Interesting thoughts but maybe not accurate. My impressions is that existing literature has focused much more on landscape hillslope - riparian heterogeneity. As I mentioned earlier, papers in hydrology and ecology have emphasized a lot on hydrological connectivity from hill to streams. In fact, the management practices related to riparian zones originated from our understanding of differences between hill and riparian and their connectivity.

As Referee #1 and #2 already pointed out, there exist several studies that cover lateral variability. Therefore we will change this sentence as described in Referee #1, L49.

Figure 3: also draw doc in this figure to help viewing when doc coming out most?

We agree, DOC measurements (stream routine and event auto samples) will be added to Figure 3.

Figure 4: this figure is busy. What is ns, hc, oc, … please explain in caption or provide legend. Why not show doc vs depth data. It would be cool to see that data. We rarely have subsurface solute depth profile. Also, these depth data, together with the modeling work for subsurface flow, provide rare opportunity to assess the relative importance of vertical heterogeneity vs horizontal landscape heterogeneity, as I I mentioned earlier

We will add the requested abbreviations to the captions.
Regarding the depth data, we are a bit in limbo. It is clear that this referee has a keen interest in the vertical distribution of DOC transport (see Figures below for DOC depth profile), but because of the nature of the soil and the bedrock, we were unable to explore this in the field in a meaningful way, as explained above. In principle, it is feasible to interrogate the model results to tease out the variations of DOC movement along the vertical dimensions but it has to be understood that the usual limitations to detailed interpretations of modeling results apply: the model is elaborate and fully 3D, but it remains a schematization of the real world, and we are not sure that an analysis of the numerical results at the level of detail desired by the referee is justifiable.

Furthermore, a discussion of the fine details of the model results will distract from the more practical aspects that referee 1 requested us to address. To us, these appear to be of more interest to the HESS readership.

[Figure]

Figure: Boxplots of DOC concentration vs. sampling depth (negative is below ground). Numbers in boxplots indicate sample size.

[Figure]

Figure: Plot of DOC concentration vs. sampling depth. Surface pond/soil solute sampling depth was set to 0 cm.

Figure 7: can the discharge data be added here? Would it be easier to understand the time series of doc export?

We agree, discharge data will be added to Figure 7.

Line 525-530: it seems that there is some mis-understanding about "lateral export". Lateral export means doc export via surface water (streams and rivers). Stream water can come from the surface runoff and subsurface (soil + gw).

We apologize for the misunderstanding, we will change the sentence to

"In this regard, we found that surficial DOC export dominated overall DOC export to the stream in our study site. "

In fact, in many places, stream water comes from "old" water from the subsurface, not surface runoff "new" water (Klaus+McDonnell 2013). While I agree that surface runoff can be important during events, it may be misleading to present these numbers without mentioning the temporal scale (event scale). At the annual scale, these numbers might be quite different.

At the annual scale, the median contribution of surficial runoff to total runoff generation was 61 % ($\pm$ 12 % standard deviation), but at the event scale surface contributions increased up to 99 % during event situations (L424, Figures 6 and 7 in the MS). Note that the stream runoff is erratic (Q ranges between < 0.01 in dry summer and > 1.1 m³ s$^{-1}$ in wet winter). Most of the runoff generation in our study site occurs during events under wet, saturated conditions. Furthermore the riparian study site has an overall low topographic relief and consists of hydromorphic soils of typically low hydraulic conductivity. The high groundwater level and the soil properties both increase the probability of surface runoff generation under the given conditions.
The generated surface runoff might be mixing of "old" water that exfiltrates from the subsurface (as indicated in our study and suggested by Frei et al. (2012)) with "new" water, but there is no isotope data or StoreAge Selection function (SAS) at hand to investigate the age distribution or preferred selection of the generated runoff. Klaus and McDonnell (2013) arrived at a different conclusion because their spatial perspective was very different from ours (see above), and possibly they were interested in much larger spatial scales. We further want to note again that the range of surface contributions ("new" water) in our catchment is in line with studies mentioned in Klaus and McDonnell (2013).

We will contrast and discuss the different conclusions from Klaus and McDonnel in the MS accordingly and emphasize the different temporal scales that we consider.

References:

Herndon, E.M., Dere, A.L., Sullivan, P.L., Norris, D., Reynolds, B. and Brantley, S.L. (2015), Landscape heterogeneity drives contrasting concentration-discharge relationships in shale headwater catchments, Hydrology and Earth System Sciences, 19(8), 3333-3347.

Jencso, K.G., McGlynn, B.L., Gooseff, M.N., Wondzell, S.M., Bencala, K.E. and Marshall, L.A. (2009), Hydrologic connectivity between landscapes and streams: Transferring reach- and plot-scale understanding to the catchment scale, Water Resour. Res., 45(4), W04428.

Klaus, J. and McDonnell, J.J. (2013), Hydrograph separation using stable isotopes: Review and evaluation, Journal of Hydrology, 505, 47-64.

Ledesma, J.L.J., Kothawala, D.N., Bastviken, P., Maehder, S., Grabs, T. and Futter, M.N. (2018), Stream Dissolved Organic Matter Composition Reflects the Riparian Zone, Not Upslope Soils in Boreal Forest Headwaters, Water Resour. Res., 54(6), 3896-3912.

McGuire, K.J. and McDonnell, J.J. (2010), Hydrological connectivity of hillslopes and streams: Characteristic time scales and nonlinearities, Water Resour. Res., 46(10).

Pacific, V.J., Jencso, K.G. and McGlynn, B.L. (2010), Variable flushing mechanisms and landscape structure control stream DOC export during snowmelt in a set of nested catchments, Biogeochemistry, 99(1-3), 193-211.

Seibert, J., Grabs, T., Köhler, S., Laudon, H., Winterdahl, M. and Bishop, K. (2009), Linking soil- and stream-water chemistry based on a Riparian Flow-Concentration Integration Model, Hydrology and earth system sciences, 13(12), 2287-2297.

Zhi, W. and Li, L. (2020), The Shallow and Deep Hypothesis: Subsurface Vertical Chemical Contrasts Shape Nitrate Export Patterns from Different Land Uses, Environmental Science & Technology, 54(19), 11915-11928.

Zhi, W., Li, L., Dong, W., Brown, W., Kaye, J., Steefel, C. and Williams, K.H. (2019), Distinct Source Water Chemistry Shapes Contrasting Concentration-Discharge Patterns, Water Resour. Res., 55(5), 4233-4251.

Additional References:

Bernhardt, E. S., Blaszczak, J. R., Ficken, C. D., Fork, M. L., Kaiser, K. E., and Seybold, E. C.: Control Points in Ecosystems: Moving Beyond the Hot Spot Hot Moment Concept, Ecosystems, 20, 665-682, 10.1007/s10021-016-0103-y, 2017.

Frei, S., Knorr, K. H., Peiffer, S., and Fleckenstein, J. H.: Surface micro-topography causes hot spots of biogeochemical activity in wetland systems: A virtual modeling experiment, Journal of Geophysical Research-Biogeosciences, 117, n/a-n/a, 10.1029/2012jg002012, 2012.

---

## Author Response (AR1)

Response to Referee 1

This study is an impressive assemblage of field, laboratory and modelling techniques to determine the spatio-temporal variability in DOC export (concentration and molecular composition) in a riparian zone. The abundance of different techniques make the manuscript quite dense and it is sometimes difficult to follow the details of the Material and methods, but I would not recommend providing more technicalities (see some exceptions in the detailed suggestions below).

My main suggestion to improve the paper is to rework the introduction and the discussion to 1) identify a clear research question or hypothesis, the introduction is lacking a "problem to solve". It was not clear to me why such a detailed study would improve management, because the resolution is far higher than any management action, or large scale modelling; 2) develop a discussion that put the results into perspective, while the current discussion is still very similar to the result section and does not contain implications for future research or management.

We appreciate your constructive, detailed evaluation of our Manuscript (MS). We realized that our research question/hypothesis was not formulated clearly enough, which is also reflected in the Reviewers statement that there is a lack of a "problem to solve" in the introduction. This was changed and addressed in the introduction and discussion sections of the MS. In line with the proposal of Referee #1 (comment on L80), we outlined our main hypothesis, that both DOC production and transport are predominantly controlled by the micro-topography of the RZ (lateral variability), and by the depth of the riparian groundwater level (temporal variability). , more precisely. This hypothesis was tested by a combination of field measurements and detailed hydrological modelling. We furthermore thoroughly reworked the introduction, discussion and conclusion with regard to implications for management and put our research into a broader perspective of literature (e.g. references and discussions given in the review of Referee #2, see also general comment on discussion of Referee #1).

The extensive rewriting removed several segments on which the reviewers commented, tendering these comments moot. We therefore cannot provide a detailed explanation about our response to these comments. In these cases it is implicitly understood that the detailed comments were addressed by the rewriting of enter sections of the paper.

I also found that the authors made too little use of the different sampling dates, especially those during storm events. I did not understand why several analyses in the manuscript only consider April and December dates, while many other dates are available (I may have missed something here…). Similarly, a high-temporal resolution sampling was performed during selected storm event but the infra-storm events dynamics is not described.

R1GC2a, b: Regarding the riparian water sampling, only 2 more samples are available for July (cf. Figure 3b). Generally groundwater sampling in summer turned out to be difficult due to low groundwater levels. Most wells, especially those screened closer to the surface, were dry in summer. To ensure proper comparability, we decided to focus on April and December, when groundwater        and        surface        water        sampling        was        possible.

Regarding the high resolution event sampling, all samples were used (cf. Figure 4a), but inter-event variance of DOC properties is higher than intra-event variance. Therefore we considered bulk sample properties of one event to be satisfactory in information content. We stated this more explicitly in the MS.

Some work would be necessary to improve the clarity of the text: shorter sentence, less and better use of conjunctions, correct some poor phrasing. I have identified a few examples of sentences to improve but note that English is not my native language either.

We worked through the MS to identify unclear or long sentences similar to those identified by the Referee and changed them accordingly.

Detailed comments:

Please note that, due to the Referees suggestions, we entirely reworked the introduction, discussion and conclusion sections. Therefore answering some of the Referees detailed comments can deviate from the original response.

Title : "from a riparian zone of a" -> "from the riparian zone of a"?

R1C1: We agree, the title was changed to

Small-scale topography explains patterns and dynamics of dissolved organic carbon exports from the riparian zone of a temperate, forested catchment

L12 "but poorly understood component": what specifically is not understood. Identify a "problem to solve", a research question or hypothesis in the abstract.

R1C2: We agree, the abstract was changed to:

Export of dissolved organic carbon (DOC) from riparian zones (RZs) is an important component of temperate catchment carbon budgets, but export mechanisms are still poorly understood. Here we hypothesize that a spatially highly resolved topographic analysis of RZs allows to characterize and delineate DOC source zones for catchment scale DOC exports

L15 "high spatio temporal resolution": what is the resolution of the DEM?

This information is given in the following sentence (1m). Therefore this sentence will not be adapted.

L15 "Stream water DOC samples from differing hydrological situations": describe these situations, number of sampling dates, study period, etc in the abstract.

R1C3a,b,c: We agree, the information was implemented in the abstract.

L18 "were then simulated": avoid passive voice throughout the manuscript

R1C4: We agree, the passive voice was changed to active where appropriate.

L20 "two distinct DOC pools (DOCI and DOCII)": describe what make them different in the abstract

R1C5: We agree, the sentence was changed to

The chemical classification by Fourier-transform ion cyclotron resonance mass spectrometry revealed revealed an aromatic, oxygen-rich DOC pool with high concentrations ($DOC_I$) and a microbially processed, mobile DOC pool with lower concentrations in the riparian groundwater and surface water samples ($n = 66$).

L22 "high-resolution topographical wetness index (TWIHR)": specify resolution in abstract.

R1C6: We agree, the resolution was specified accordingly.

L27 "should be considered in DOC export models": any implications for management? Should large-scale models really consider this fine-resolution heterogeneity?

Considering fine-resolution heterogeneity is likely not applicable to large-scale models, but understanding the export mechanisms of riparian zones at fine scale allows to better estimate overall DOC export potential of catchments as a function of climatic variability and general topographic structure. However, the relationship between riparian zone structural heterogeneity and DOC export presented in this study suggests that knowledge about riparian topography and structure, easily derived from DEMs, may be useful for the development of more parsimonious models for the prediction of hydrologic and DOC export response by e.g. implementing a threshold-based surface runoff module. Measures of riparian zone source connectivity (like the presented TWIHR) provide an integrated measure of riparian zone surface runoff generation and the associated DOC export behavior that is – when integrated appropriately – scalable to catchment level. We therefore believe that proxies of fine-resolution structural heterogeneity can improve large-scale catchment models, which cannot represent topographic (riparian) relief at very fine scale. We agree with the referee that our detailed, fine-resolution modelling effort will generally not be feasible for day-to-day management operations. However, it helped us to demonstrate the usefulness of drone-based DEMs to gain process understanding. With the help of the model we could illustrate the importance of the TWIHR for DOC export.

R1C7: We addressed this in the Introduction and Discussion (section 4.4) and Conclusions of the MS.

L27 "But despite": don't start a sentence with "but"

Please see next comment (R1C8).

L33 "but could" second but in this long sentence

R1C8: We agree, the sentence was changed to

Changes in land use, climate and biogeochemical boundary conditions have increased DOC concentrations in surface waters and altered the quality of the exported DOC in the last decades (Larsen et al., 2011; Chantigny, 2003; Wilson and Xenopoulos, 2008). Beside the ecological impacts this alteration may also affect safety and costs of drinking water production (e.g. Wang et al., 2017). Routine management of DOC could therefore help to comply with water quality directives and lower the cost of drinking water purification (Matilainen et al., 2011), but is currently almost non-existent (Stanley et al., 2012).

L36 "Especially riparian zones (RZs) of lower order streams are potential targets for…" poor phrasing

R1C9: We agree, the sentence was changed to

Lower order streams make up a large fraction of the total river networks worldwide (Raymond et al., 2013) and their riparian zones (RZs) represent a main source for terrestrial DOC export (Ledesma et al., 2015; Musolff et al., 2018). Therefore RZs of lower order streams – as terrestrial-aquatic interfaces – constitute a general control unit, qualifying them as potential targets for DOC export management.

L41 "Here, DOC …" add reference

R1C10: We agree, the following reference was added:

Luke, S. H., Luckai, N. J., Burke, J. M., and Prepas, E. E.: Riparian areas in the Canadian boreal forest and linkages with water quality in streams, Environmental Reviews, 15, 79-97, 10.1139/A07-001, 2007.

L46 "This leads to a stronger accumulation of DOC close to the soil surface…" I did not understand the link with the previous sentence

R1C11: We agree, the sentence was changed to

On the other hand, the amount of accumulated DOC and ultimately export is also dependent on hydrological connection of DOC sources to the stream, because water can only mobilize DOC pools if these contribute to riparian runoff generation.

L49 "led to concepts like variable source zone activation (Dick et al., 2015; Werner et al., 2019), the dominant source layer (Ledesma et al., 2015) and transmissivity feedback (Bishop et al., 2004)" explain these concepts and their limits. Listing them is not enough in an introduction

R1C12: The concepts were explained. We reworked this paragraph to

Several attempts have been made to characterize and quantify the dynamics of runoff generation in RZs and the associated variability of DOC transport to streams. However, to date model conceptualizations have mainly focused on the vertical distribution of DOC sources in the subsurface and to a lesser degree on horizontal heterogeneity induced by topography. For

instance the dominant source layer concept (Ledesma et al., 2015) focuses on depth-dependent differences in DOC pools in distinct soil layers, which are assumed to be uniform across the RZ. Transmissivity feedback (Bishop et al., 2004) accounts for depth-dependent differences in hydraulic conductivities of soils and the resulting changes in the transmissivity of the soil profile under changing groundwater levels. This concept is taken up in the riparian profile flow-concentration integration model (Rim, Seibert et al., 2009) to model stream solute variability as a function of a non-linear vertical distribution of pore water solute concentrations in riparian soils. Frei et al. (2010, 2012) were able to simulate the complex effects of riparian micro-topography on runoff generation and the formation of biogeochemical hotspots in the subsurface, but their explorative model was computationally expensive and did not explicitly consider DOC transport. Variations in the lateral hydrological connectivity of a RZ to a stream have been conceptualized in a spatially lumped catchment DOC export model by defining different source zones with variable activation (Dick et al., 2015). In essence, most model conceptualizations for DOC export from RZs describe a heterogeneous system in terms of spatially lumped integrated functional relationships without explicitly acknowledging small-scale spatio-temporal variability in DOC export from individual, small landscape units (Ledesma et al., 2018a; Dick et al., 2015).

L55 "a strong focus on vertical heterogeneity" I my understanding the variable source area concept is more about horizontal heterogeneity.

R1C13: We agree, the text was changed to (please also see comment above)

Variations in the lateral hydrological connectivity of a RZ to a stream have been conceptualized in a spatially lumped catchment DOC export model by defining different source zones with variable activation (Dick et al., 2015).

L57 "Moreover RZs are highly dynamic and heterogeneous with micro-topography" the role of micro topography is central to the hypothesis of this work and should be better highlighted.

R1C14: We agree, the role of micro-topography was better highlighted. We changed the text to

Micro-topography in RZs can induce hot spots of biogeochemical activity (Frei et al., 2012) that contribute disproportionally to nutrient turnover. Temporary, hot spots of DOC production in the shallow soil layers of the RZ can become hydrologically connected to the stream (during hot moments), when the groundwater levels intercept the surface and micro-topography focuses drainage (Frei et al. 2010, Scheliga et al., 2019) and can consequently be a source for solute exports to the stream. Therefore, micro-topography in the RZ is considered a fundamental organizing structure, not only for soil chemistry (Diamond et al., 2020) but also of hydrological connectivity (Frei et al. 2010, Scheliga et al., 2019) that induces high spatio-temporal heterogeneity of DOC exports. Riparian topography and the dynamics of groundwater levels in the RZ thus are the key drivers of the spatio-temporal patterns of DOC export from RZs.

Additional references:

Diamond, J. S., McLaughlin, D. L., Slesak, R. A., and Stovall, A.: Microtopography is a fundamental organizing structure of vegetation and soil chemistry in black ash wetlands, Biogeosciences, 17, 901-915, 10.5194/bg-17-901-2020, 2020.

Scheliga, B., Tetzlaff, D., Nuetzmann, G., and Soulsby, C.: Assessing runoff generation in riparian wetlands: monitoring groundwater–surface water dynamics at the micro-catchment scale, Environmental Monitoring and Assessment, 191, 116, 10.1007/s10661-019-7237-2, 2019.

L65 "Model conceptualizations that are able to bridge those scales" with this sentence it seems that the paper will deal with this question of scales, but it is not the case.

R1C15: We agree, the sentence was removed from the manuscript.

L80 "We argue that a smaller-scale, dynamic assessment of the TWI…" should be the hypothesis of the paper. Please give a response to his hypothesis/question in the discussion/conclusion.

R1C16: We agree, in our reworked MS, we hypothesize that both DOC production and transport are predominantly controlled by the micro-topography of the RZ (lateral variability), and by the depth of the riparian groundwater level (temporal variability).The hypothesis was tested by a combination of field measurements and detailed hydrological modelling. The discussion and conclusion were adapted accordingly to respond to this hypothesis.

L97 "In this paper we…" intro long enough, no need for a summary of the methods here. Develop problem to solve instead.

R1C17a: We agree, we reworked the entire paragraph. The first methodological parts of this paragraph were deleted and instead replaced by a problem to solve.

L100 "More specifically, (1)…" it would be better to list specific research questions than summary of the methods.

R1C17b: We agree, please see comment above (R1C17a).

L128 "Electric resistivity tomography (Resecs DC resistivity meter system, Kiel, Germany) was applied at two transects" show the transects in figure 1?

R1C18a: We agree, transects of the conducted electric resistivity tomography are shown in Figure 1b. The captions were adapted accordingly.

L134 "Two PCM4 portable flow meters (Nivus, Germany) measured discharge in the Rappbode stream at a chosen inlet…" show inlet and outlet in figure 1?

R1C18b: We agree, locations of in- and outlet are shown in Figure 1b.

L151 "To have maximum ability in capturing the magnitude and direction of this slope…" poor phrasing

R1C19: We agree, the sentence was rephrased to

We installed a piezometer network aligned on a square grid, with one principal axis oriented in parallel to the stream and the other perpendicular to the stream to capture the temporal dynamics of the groundwater level in both principal directions of this slope.

L156 "In addition 3 more wells were installed at 0.3 m depth inside the rectangular grid for surface near sampling." I did not understand this sentence.

R1C20a: We agree, we changed the sentence to

In addition we irregularly installed three wells with screens at 0.3 m depth (but no pressure transducers) inside the piezometer network for sampling near the surface.

2.2.3. I found it difficult understand the maximum depth and the screening height of the different piezometers and wells. Please rework this section to improve clarity.

R1C20b: We agree, the section was reworked with focus on improved clarity.

L166 "Biweekly routine samples…" it is never clear whether biweekly means twice a week or every second week. Please use a less ambiguous term. Please also add the number of sampling dates and the number of dates when FT-ICR mass spectrometry was used. Is it only two dates?

R1C21: We agree, the number of sampling dates was added. We further realized that the referee was confused about the description of FTICRMS samples. We therefore reworked the entire paragraph.

L181 "samples were filtered using 0.45 μm membrane filters" did you filter the samples in the field or back in the lab?

R1C22: The sentence was changed to

Samples were filtered (0.45 μm membrane cellulose acetate filters, rinsed with 20 mL of sample water to avoid bleeding; Th. Geyer, Germany) and acidified to pH 2 (HCl, 30 %, Merk, Germany) on site. Subsequently samples were stored cool (4 °C) and dark until timely DOC measurement and extraction in the laboratory.

L280 "2.4.3 Calibration" is it possible to provide the objective function of the calibration? I understood that the model aimed to simulate both the stream discharge and groundwater depth in several wells, with a weighting scheme giving a high importance to the groundwater, but it would be interesting to see the equation of this objective function.

R1C23: We agree, we clarified this in the MS and provided the objective function in the SI (S2)

$$\text{Multi} - \text{Objective function} = \sum_{i=1}^{i=nq} w_q (O_q^i - S_q^i)^2 + \sum_{i=1}^{i=nl} w_l (O_l^i - S_l^i)^2$$

Where $O_q^i$ and $S_q^i$ are the observed and simulated discharge. $nq$ is the number of number of the discharge observations (611). $O_l^i$ and $S_l^i$ are the observed and simulated groundwater level. $nl$ is the number of groundwater level observations (110140). $w_q$ and $w_l$ are the weights for the two observation groups, both being assigned with the value of 1 in the calibration. Because the observation number of groundwater level was significantly larger than that of the discharge, this multi-objective function highlight the importance of the groundwater levels, such that the 94% of the multi-objective function for the calibrated best-fit was attributed to groundwater levels.

L322 "The DInf algorithm was used" please explain what it is.

R1C24: The DInf algorithm is explained in the revision.

We applied the DInf algorithm to calculate a realistic hydrological routing (Tarboton, 1997). The DInf algorithm determines flow direction as the steepest downward slope on eight triangular facets formed in a 3x3 cell window centered on the cell of interest.

L335 "Discharge shows event-type, erratic variability" poor phrasing

R1C25: We agree, the sentence was changed to

Discharge showed high variability at the event-scale. At annual scale, discharge expressed a clear seasonal pattern, with lowest values in late summer and highest values in spring (Fig. 3a).

L360 "DOC in riparian water samples was in general of highly unsaturated and phenolic composition, typically found in lignin and biomass type compounds" can we see this in a table or a figure?

R1C26: This can be derived from Figure S10. We have noticed that the colors in this Figure were wrong and included a corrected version of Figure S10:

[Figure]

**Figure S10: Aggregated van Krevelen plot of all FT-ICR-MS sample of stream (blue) and riparian origin with type DOCI (grey) and DOCII (yellow). Data represent the intensity weighted average (wa) of the molecular H/C and O/C ratios considering all valid MF in these samples (n = 142). See also Table S5 for individual values.**

Besides, the sentence in L360 (R1C26) was changed to

In general, DOC in riparian water samples was of unsaturated and phenolic composition (waHC = 1.27 ± 0.05; waOC = 0.40 ± 0.01; n = 66), that is typically found in wetland surface soils (LaCroix 2019). Stream event samples significantly differed (p < 0.001) from riparian samples and were more unsaturated (waHC = 1.17 ± 0.05; n = 76) and more oxygenated (waOC = 0.43 ± 0.03) as shown in Fig. S10.

L395 "Note that wells, sampled during different occasions throughout the year occur in both DOC clusters and according TWIHR values can thus occur in both clusters" it is unclear to me to what extend a given piezometer belonged to the same cluster throughout time. This sentence suggests that the cluster can change, but a quantitative assessment of how many piezometer remain in the same cluster or change clusters would be interesting here.

R1C27: We agree, a quantitative assessment of how many piezometer remain in the same cluster or change clusters was added:

Note that 7 (i.e. 6 wells and one surface pond sample) out of 15 locations occur in both DOC clusters as DOC quality varied over time. According TWI$_{HR}$ values also contribute to both clusters (Fig. 4d)

L415 "The significant difference in TWIHR median values of DOCI and DOCII wells" I did not understand how you could classify wells as DOCI-well or DOCII-well if a given well could change clusters in different dates.

R1C28: We agree, the correct term in this sentence is DOCI and DOC II *clusters*, not *wells*. We changed this in the manuscript and apologize for the inconvenience.

L416 "using the median TWIHR value of the DOCI group (9.66) as a threshold." I did not understand this choice; please explain the rationale behind this.

R1C29: The rationale was to map DOC source zones of different DOC concentration and chemical properties. We found an overlap of TWIHR values between both groups although their median was found to be significantly different. Using the median of DOCI (9.66) as a manually chosen threshold we can separate both groups capturing 50% of all cases of group I in one class while only allowing 25% of group II.

We realized that the rationale behind the selection of the threshold was not addressed clearly enough in the MS. Therefore we changed the sentence to… by using the median $TWI_{HR}$ value of the well locations of the $DOC_I$ cluster (9.66) as a threshold. Using this manually chosen threshold allowed to allocate the samples of both DOC clusters to two distinct $TWI_{HR}$-based groups. In this way, more than 50% of samples contributing to the $DOC_I$ cluster constitute one group while allowing less than 25% (15% in April) of the samples contributing to the $DOC_{II}$ cluster in that group

L418 "Also note that different samples of one well can appear in both DOC groups" please give numbers.

R1C30: We agree, numbers were added in the manuscript.

L435 "Fig. S7, Table S4 for according water fluxes" -> "corresponding water fluxes"?

R1C31: We agree, this was changed in the manuscript.

L454 "During the model period, DOCI source wells had a median DOC concentration of 5.8 mg L-1 which was 2.3 times higher than for the DOCII source wells" it would be interesting to remind the mean+/- sd of the two types of wells. Do deeper wells match with the DOCI cluster?

R1C32: We agree, ± sd was added:

During the model period, $DOC_I$ source wells had a median DOC concentration of 5.8 mg $L^{-1}$ (mean ± SD: 6.2 ± 2.7 mg $L^{-1}$), which was 2.3 times higher than the median for the $DOC_{II}$ source wells (2.7 ± 1.2 mg $L^{-1}$).

Certain deeper wells match with the DOCI cluster (e.g. A1-E1, B4, C4, see also comment below).

L487 "as typically found in deeper soil layers" what influences the difference between DOC I and DOC II more: the TWI or the sampling depth? (or both are related?).

R1C33: There is no statistical significant relation between sampling depth and well classification in our samples (see Figure below). On the other hand, we presented significant differences in TWI$_{HR}$ between the clusters. We therefore conclude that TWI (as postulated in the MS) controls the difference between DOCI and DOC II (more).

However, the possibility of a bias exists since samples were predominantly taken in deeper soil layers - also due to the fact that there often was no surface near water available when groundwater samples were taken. Moreover, surface near samples as well as deep samples appear in both clusters.

Also with regard to referee #2 (R2C4a, b) we included this discussion in the MS..

[Figure]

Figure: Boxplots of piezometer slot/sampling depth [cm below ground] and TWI [-] value for the DOC$_I$ and DOC$_{II}$ clusters. Horizontal brackets above describe the Kolmogorov-Smirnoff (KS) and Wilcoxon rank sum (WC) test statistics. Values were min-max normalized to values between 0 and 1 for better illustration.

L491 "indicating a replete DOC pool with constant contribution to the overall DOC quality in the stream" unclear sentence

R1C34: We agree, the sentence was changed to

In addition, the $DOC_I$ quality was similar between April and December indicating a DOC pool which is not strongly affected by seasonality and hydrologic conditions (i.e. steady export during the wet and cold state). Therefore $DOC_I$ can be regarded as a permanently available source of stream water DOC.

L493 "indicating the influence of seasonality on this pool." It is difficult to make such a conclusion with only two dates.

R1C35: We agree, the sentence was changed to

In contrast, the $DOC_{II}$ composition was reflected in the DOC composition of stream water in December but not in April, suggesting a connectivity of this pool during high flow periods but a potential depletion over time.

General comment on "4 Discussion": this discussion is too similar to the result section, many conclusions are specific to the study site while readers would expect to see the results put into perspective, with more implications for management and research, more key messages and more references to the literature.

R1GC on Discussion: We agree. In consistence with Referee #2 we reworked the introduction, discussion and conclusions section (see general comment and comments on L12, L27). Here primary focus lied on the following tasks:

- The new hypothesis "We therefore hypothesize that both DOC production and transport are predominantly controlled by the micro-topography of the RZ (lateral variability), and by the depth of the riparian groundwater level (temporal variability)" was carefully established, discussed and ultimately answered and more clear key messages were worked out.
- We ensured that drawn conclusions were applicable to similarly shaped riparian zones.
- Further, results were better put into perspective and referenced to the literature e.g. by answering Referee #2's comments on horizontal vs. vertical heterogeneity (R2C4a).
- We added a new discussion section "4.4 Potential for future work and implications" and a paragraph in the conclusions to better promote implications for management and research.

Werner et al. examines export patterns and dynamics of dissolved organic carbon from the riparian zone in a temperate, forested catchment. The paper used an array of different approaches to relate DOC source zones within the RZ to their dominant DOC export mechanisms. Stream DOC samples from different hydrological conditions were compared to riparian DOC groundwater and surface water chemistry. They also characterized DOC chemically (via Fourier-transform ion cyclotron resonance mass spectrometry) and used topographic analysis (at a resolution of 1m). Water fluxes were simulated using the code HydroGeoSphere. The paper concluded that surface runoff from zones of high TWIHR values, which occupied about 15% of the total area, exported about 1.5 times the load of DOC from the remaining 85 % of the area, and that "this study highlights that surface DOC export from the riparian zone plays an important role for lateral DOC export from hydromorphic soils with overall low topographic relief."

The work is interesting and collated an array of approaches from chemical analysis to modeling at high spatial resolution. The current manuscript is phrased from the angle of horizontal heterogeneity / landscape topography, which I think is actually already well studied [Herndon et al., 2015; Jencso et al., 2009; Ledesma et al., 2018; McGuire and McDonnell, 2010; Pacific et al., 2010]. But I find the conclusion is not particularly surprising.
On the other hand, it seems to me that this work presents a rare opportunity to dig deeper to think about the relative influence of vertical versus horizontal heterogeneity. The relative importance of vertical versus horizontal heterogeneity in doc export is poorly understood. In particular, there has been quite some interests in understanding the solute export from different subsurface depths, for example, [Seibert et al., 2009; Zhi and Li, 2020; Zhi et al., 2019]

We appreciate your evaluation of our Manuscript (MS). Your comments as well as those by the other Referee made it clear that we needed to phrase our conclusions more crisply and clarify the contribution of our work to the existing body of knowledge and its practical implications (see also Referee #1, General Comment on "4 Discussion"). We discussed the references you mention and put our results into perspective (see section 4.3, 4.4 and R2C5).

Please note that we reworked the introduction, discussion and conclusion sections to a greater extent. Therefore it was to some (small) degree not possible to answer the Referees comments explicitly anymore. In those cases, the questions and remarks were implicitly addressed in the rewritten text.

R2GC1: Please note that most of the references you mention (Herndon et al., 2015; McGuire and McDonnell, 2010; Klaus and McDonnell, 2013; Zhi and Li, 2020; Zhi et al., 2019) are concerned with much larger scales than our work, which studies an individual riparian zone (RZ) in detail. The scale of our experimental site is the smallest on which a hydrological landscape unit (in this case, the RZ) can be studied in its entirety and the largest that still allows extensive monitoring in the field during several seasons within realistic constraints on resources and personnel. The paper's contribution lies in the combination of multi-sensor field monitoring backed up by detailed hydrological modelling and high-resolution chemical analyses of DOC quantity and quality. This allowed us to determine the size and contribution of hydrologically different DOC source areas, something that so far has only rarely been quantified (Bernhardt et

al., 2017). We better clarified this in the text (section 4.3) and improved the explanation of the practical relevance of some of our findings (section 4.4).

The relative influence of vertical versus horizontal heterogeneity was discussed in our MS (R2GC1, see also Referee #1, L487). As mentioned above, we found that surficial DOC export from high TWI zones dominates DOC export from riparian zones to the stream at our study site suggesting that horizontal heterogeneity predominates vertical heterogeneity in low relief catchments with hydromorphic soils. With our work showing that surface flow is an important carrier of DOC stemming from localized source areas, it contrasts the concepts of a dominant source layer in the subsurface (Ledesma et al., 2018) or a riparian integration model (Rim, Seibert et al., 2009) that both hypothesize predominantly horizontal subsurface flow as the main transport mechanism for DOC. This suggests – as also assumed by Jensco et al., 2009 – a dependency of DOC export on morphologic, climatic and topographic conditions. Given the nature of our study site (steep hillslopes, level riparian zone, low conductivity soils), overland flow is to be expected as a significant export pathway connecting riparian DOC sources to the stream, whereas subsurface DOC export heterogeneity seems to be relatively unimportant. We discussed this in section 4.2 and 4.3 (R2C5 and R2GC1, resp.).

The data from this work have depth profile (top 100 cm) of doc, and flow calcination from different depths. These two can be combined to calculate at what depth most doc was exported, and how the export varied with depth in high flow events. At a minimum, it would be nice to see some discussion along this line of vertical heterogeneity.

We agree, some discussion along this line has been added to the MS (see also: comments below, specific response to comment of Referee #1, L487, comment above). We point out that the top few decimeters of soil was very organic and contained many rhizomes. Somewhat deeper, the soil contained so many rock fragments that digging with a spade was no longer possible. At roughly 1 m (with considerable variations), fractured bedrock occurred. To study solute export from various depths in the field, tracer experiments are needed. However, taking samples at well-defined locations in this soil proved very difficult, even in the dry season when the groundwater was not near the soil surface. We do not believe it is feasible to excavate the soil in intervals of 5 or 10 cm to study the distribution of a dye tracer. Given the difficulties associated with tracer experiments to identify flow paths in this soil that we outlined here, this discussion was based entirely on the numerical model and therefore has to rely on a schematized conceptualization of the subsurface.

I also find "Surface export" is a confusing term. Is this really surface runoff, or does the water mostly flow through top soil? Unless in extremely large events, most forests do not see significant amount of surface runoff. In many places, stream water comes from "old" water from the subsurface, not surface runoff "new" water [Klaus and McDonnell, 2013].

R2GC2: There is no ambiguity in our use of the term. The RZ on which this paper focuses was not forested, although the surrounding slopes were. Overland flow in the RZ was quite common during wet periods (R2C5). With "surface export" we refer to water that has been on the surface at least once on its way to the stream – as indicated by respective exchange fluxes. We clarified

this in the text (R2GC2). Surface flow can reinfiltrate and flow to the stream/ boundaries through the subsurface.

The steep hillslope but mild slope and hydromorphic soil in the study site, the micro-topography, as well as scale and climatic setting of our field site all are in stark contrast with the features of the sites reviewed by Klaus and McDonnell (2013). Also, they view water contributions from a watershed/catchment-scale perspective, whereas we focus on the RZ itself. Given the nature of our study site, more overland flow is to be expected (although Klaus and McDonell also reviewed studies that had up to 100% "new" water contributions). We explicitly mentioned that our findings hold for a low relief riparian zone with hydromorphic soils to account for these differences (R2C5).

As discussed below (comment on L525 and following) we therefore do not think that the concept presented in Klaus and McDonnell holds for our field site.

Line 52-54, "a strong focus on vertical heterogeneity". Interesting thoughts but maybe not accurate. My impressions is that existing literature has focused much more on landscape hillslope - riparian heterogeneity. As I mentioned earlier, papers in hydrology and ecology have emphasized a lot on hydrological connectivity from hill to streams. In fact, the management practices related to riparian zones originated from our understanding of differences between hill and riparian and their connectivity.

Referee #1 and #2 both pointed to several studies that cover lateral variability. Therefore we changed this sentence as described in our response to Referee #1, L49 (R1C12&13).

Figure 3: also draw doc in this figure to help viewing when doc coming out most?

R2C1: We agree, DOC measurements (riparian, stream routine and event auto samples) were added to Figure 3.

Figure 4: this figure is busy. What is ns, hc, oc, … please explain in caption or provide legend. Why not show doc vs depth data. It would be cool to see that data. We rarely have subsurface solute depth profile. Also, these depth data, together with the modeling work for subsurface flow, provide rare opportunity to assess the relative importance of vertical heterogeneity vs horizontal landscape heterogeneity, as I I mentioned earlier

R2C2: We added the requested abbreviations to the captions.

Regarding the depth data, we are a bit in limbo. It is clear that this referee has a keen interest in the vertical distribution of DOC transport (see Figures below for DOC depth profile), but because of the nature of the soil and the bedrock, we were unable to explore this in the field in a meaningful way, as explained above. In principle, it is feasible to interrogate the model results to tease out the variations of DOC movement along the vertical dimensions but it has to be understood that the usual limitations to detailed interpretations of modeling results apply: the model is elaborate and fully 3D, but it remains a schematization of the real world, and we are not sure that an analysis of the numerical results at the level of detail desired by the referee is justifiable.

Furthermore, a discussion of the fine details of the model results will distract from the more practical aspects that referee 1 requested us to address. To us, these appear to be of more interest to the HESS readership.

[Figure]

Figure: Boxplots of DOC concentration vs. sampling depth (negative is below ground). Numbers in boxplots indicate sample size.

[Figure]

Figure: Plot of DOC concentration vs. sampling depth. Surface pond/soil solute sampling depth was set to 0 cm.

Figure 7: can the discharge data be added here? Would it be easier to understand the time series of doc export?
R2C3: We agree, discharge data was added to Figure 7. Figure captions were adapted accordingly.

Line 525-530: it seems that there is some mis-understanding about "lateral export". Lateral export means doc export via surface water (streams and rivers). Stream water can come from the surface runoff and subsurface (soil + gw).

R2C4: We apologize for the misunderstanding, we changed the sentence to

In this regard, we found that surficial DOC export dominated overall DOC export to the stream at our study site.

In fact, in many places, stream water comes from "old" water from the subsurface, not surface runoff "new" water (Klaus+McDonnell 2013). While I agree that surface runoff can be important during events, it may be misleading to present these numbers without mentioning the temporal scale (event scale). At the annual scale, these numbers might be quite different.

R2C5: At the annual scale, the median contribution of surficial runoff to total runoff generation was 61 % (± 12 % standard deviation), but at the event scale surface contributions increased up to 99 % during event situations (L424, Figures 6 and 7 in the MS). Note that the stream runoff is erratic (Q ranges between < 0.01 in dry summer and > 1.1 m³ s⁻¹ in wet winter). Most of the runoff generation in our study site occurs during events under wet, saturated conditions. Furthermore the riparian study site has an overall low topographic relief and consists of hydromorphic soils of typically low hydraulic conductivity. The high groundwater level and the soil properties both increase the probability of surface runoff generation under the given conditions.

The generated surface runoff might be mixing of "old" (pre-event) water that exfiltrates from the subsurface (as indicated in our study and suggested by Frei et al. (2012)) with "new" (event) water, but there is no isotope data or StoreAge Selection function (SAS) at hand to investigate the age distribution or preferred selection of the generated runoff. Klaus and McDonnell (2013) arrived at a different conclusion because their spatial perspective was very different from ours (see above), and possibly they were interested in much larger spatial scales. We further want to note again that the range of surface contributions ("new"/event water) in our catchment is in line with studies mentioned in Klaus and McDonnell (2013).

We discussed and emphasized the different conclusions from Klaus and McDonnel in the MS accordingly.

References:
Herndon, E.M., Dere, A.L., Sullivan, P.L., Norris, D., Reynolds, B. and Brantley, S.L. (2015), Landscape heterogeneity drives contrasting concentration-discharge relationships in shale headwater catchments, Hydrology and Earth System Sciences, 19(8), 3333-3347.
Jencso, K.G., McGlynn, B.L., Gooseff, M.N., Wondzell, S.M., Bencala, K.E. and Marshall, L.A. (2009), Hydrologic connectivity between landscapes and streams: Transferring reach- and plot-scale understanding to the catchment scale, Water Resour. Res., 45(4), W04428.
Klaus, J. and McDonnell, J.J. (2013), Hydrograph separation using stable isotopes: Review and evaluation, Journal of Hydrology, 505, 47-64.
Ledesma, J.L.J., Kothawala, D.N., Bastviken, P., Maehder, S., Grabs, T. and Futter, M.N. (2018), Stream Dissolved Organic Matter Composition Reflects the Riparian Zone, Not Upslope Soils in Boreal Forest Headwaters, Water Resour. Res., 54(6), 3896-3912.
McGuire, K.J. and McDonnell, J.J. (2010), Hydrological connectivity of hillslopes and streams: Characteristic time scales and nonlinearities, Water Resour. Res., 46(10).
Pacific, V.J., Jencso, K.G. and McGlynn, B.L. (2010), Variable flushing mechanisms and landscape structure control stream DOC export during snowmelt in a set of nested catchments, Biogeochemistry, 99(1-3), 193-211.
Seibert, J., Grabs, T., Köhler, S., Laudon, H., Winterdahl, M. and Bishop, K. (2009), Linking soil- and stream-water chemistry based on a Riparian Flow-Concentration Integration Model, Hydrology and earth system sciences, 13(12), 2287-2297.
Zhi, W. and Li, L. (2020), The Shallow and Deep Hypothesis: Subsurface Vertical Chemical Contrasts Shape Nitrate Export Patterns from Different Land Uses, Environmental Science & Technology, 54(19), 11915-11928.
Zhi, W., Li, L., Dong, W., Brown, W., Kaye, J., Steefel, C. and Williams, K.H. (2019), Distinct Source Water Chemistry Shapes Contrasting Concentration-Discharge Patterns, Water Resour. Res., 55(5), 4233-4251.

Additional References:

Bernhardt, E. S., Blaszczak, J. R., Ficken, C. D., Fork, M. L., Kaiser, K. E., and Seybold, E. C.: Control Points in Ecosystems: Moving Beyond the Hot Spot Hot Moment Concept, Ecosystems, 20, 665-682, 10.1007/s10021-016-0103-y, 2017.
Frei, S., Knorr, K. H., Peiffer, S., and Fleckenstein, J. H.: Surface micro-topography causes hot spots of biogeochemical activity in wetland systems: A virtual modeling experiment, Journal of Geophysical Research-Biogeosciences, 117, n/a-n/a, 10.1029/2012jg002012, 2012.

---

## Referee Report (RR1)

This study uses high-resolution field sampling and surface-subsurface hydrologic modelling techniques to determine the spatial and temporal variability in DOC sources and export from a riparian zone. The authors found that two distinct clusters of DOC concentration and composition could be explained by topographic wetness index, which was then used to delineate DOC source zones within the riparian zone. DOC export from high TWI zones was 1.5 times greater than low TWI zones. Overall, this study is an impressive case study of how, when, and where DOC is exported from the riparian zone in a small headwater catchment.

The number of different field, lab, and modeling techniques employed make this manuscript difficult to follow at times. While much of this difficulty is unavoidable due to the complex nature of the research question, I have made suggestions for the authors to simplify language, particularly around descriptors of their DOC clusters, to help make the intent of their use more clear and purposeful.

As per previous reviewers suggestions, the authors have reworked the introduction and discussion to 1) identify a clear research question or hypothesis and 2) develop a discussion that put the results into perspective. The extensive effort on the author's part to address these comments is commendable and has resulted in a compelling discussion of their results and a well formed hypothesis and introduction. I agree with previous reviewer suggestions that the rationale behind how this study is relevant to management or the argument that DOC export needs to be managed is unclear. I suggest that the authors reframe the first few paragraphs of the introduction to be centered around larger knowledge gaps around linkages between terrestrial-aquatic carbon cycling, transport, and fate. For these reasons listed above, I suggest that this manuscript be accepted for publication pending minor revisions. I have included line-by-line comments below for specific areas throughout the manuscript.

L16 (Abstract): This hypothesis does not match the hypothesis in your introduction, or the hypothesis that is referenced throughout the MS.

L22 (Abstract): Should (n = 66) be ($DOC_{II}$)?

L24 (Abstract): Here and elsewhere in the abstract (and main text), "pool", "type", "source zone" and "cluster" are all used in reference to $DOC_I$ and $DOC_{II}$. These descriptors all appear to be used interchangeably, but you are 1) using a cluster analysis to isolate and contrast end members within the broader DOM pool and 2) you are using zones to refer to both DOC and TWI. Also, shouldn't "$DOC_I$ source zone with high $TWI_{HR}$ values" be "high $TWI_{HR}$ zones associated with the $DOC_I$ cluster", because the zones you are referencing were categorized by $TWI_{HR}$ and then assigned a DOC cluster based on the $TWI_{HR}$ value? I recommend that the authors simplify these descriptors throughout the abstract and MS to just DOC "clusters" to avoid confusion and be representative of the DOC comparison analyses conducted.

L34-45 (Introduction): This first introduction paragraph/section needs more detail and evidence to build the argument that DOC is important. DOC in streams and rivers is of central ecological importance to what? The argument in this paragraph does not support the claim that DOC export needs to be managed and this study does not address questions in which a "for management"

framing seems appropriate. More generally, what I think this study does do is use an impressive high resolution field and modeling approach to ask how, when, and where is DOC entering the stream from the riparian zone. DOC generation, understanding how DOM changes and moves within and across ecosystem interfaces, and linking aquatic and terrestrial carbon cycling are still large knowledge gaps that are 1) needed to then argue for DOC export management and 2) knowledge gaps that this study is addressing! I would suggest returning to the Cole et al. 2007 paper you cite to help reframe this first section of the introduction. I've also included a few citations below of recent papers to help frame this argument:

- Butman D., R. Striegl, S. Stackpoole, P. del Giorgio, Y. Prairie, D. Pilcher, P. Raymond, F. Paz Pellat, and J. Alcocer (2018), Chapter 14: Inland waters. In *Second State of the Carbon Cycle Report (SOCCR2): A Sustained Assessment Report*. U.S. Global Change Research Program, Washington, DC, USA. 568-595.
- Drake, T. W., P. A. Raymond, and R. G. M. Spencer (2018) Terrestrial carbon inputs to inland waters: A current synthesis of estimates and uncertainty. *Limnology & Oceanography Letters*, 3, 132-142.
- Vachon, D., R. A. Sponseller, and J. Karlsson (2021), Integrating carbon emission, accumulation and transport in inland waters to understand their role in the global carbon cycle. *Global Change Biology*, 27, 719-727. https://doi.org/10.1111/gcb.15448

L60-95 (Introduction): After reviewing the author's changes and comments from the last round of review, I wanted to say that this section of the introduction does an excellent job of setting up your study, why it matters, and why its important. Great job!

L170 (Methods): Leaving auto-sampled stream water unfiltered and unpreserved for up to 4 days affects both your DOC concentration and the molecular composition. Most short term assessments of biodegradable DOC last 4 days where a significant amount of DOC can be taken up (Catalan et al. 2021 found up to 40% of initial DOC could be consumed with the first 200 hours). Can you address this potential degradation effect in some way? Did you auto-sample the same well or in the stream several days in a row/between trips to collect and filter samples? This degradation effect likely affected each of your samples differently as well, depending on the time left unfiltered as well as the DOM and microbial community composition. Some relevant studies to consider:

- Catalán, N., Pastor, A., Borrego, C.M., Casas-Ruiz, J.P., Hawkes, J.A., Gutiérrez, C., von Schiller, D. and Marcé, R. (2021), The relevance of environment vs. composition on dissolved organic matter degradation in freshwaters. Limnol Oceanogr, 66: 306-320. https://doi.org/10.1002/lno.11606
- D'Andrilli, J., Junker, J.R., Smith, H.J. *et al.* DOM composition alters ecosystem function during microbial processing of isolated sources. *Biogeochemistry* **142,** 281–298 (2019). https://doi.org/10.1007/s10533-018-00534-5

L388 (Results 3.2.1): This is a clear description of the $DOC_I$ cluster, but you as need one for the $DOC_{II}$ as well. Also another reminder to be clear and purposeful with the terms used to describe your DOC clusters (this section is clear and the use of clusters is deliberate).

L435 (Results 3.2.2): This is a clear definition of $DOC_I$ and $DOC_{II}$ source zones and agree that following this point, these terms can be used. I also appreciate the parenthetic reminders in the results and discussion (i.e., "high TWI zones"). However, because this definition is buried in the results, I suggest reworking your abstract to be clear around source zones vs. clusters.

L470 (Results 3.3): Are "DOC source wells" the same as "DOC source zones"? Maybe change to "wells in DOC source zones" to be more clear?

Discussion: I wanted to commend the authors on restructuring their discussion! The discussion is distinct from the results, provides context and explanation of key findings, and stresses the importance of the work (all of which were recommendations made by previous reviewers).

L505 (Discussion 4.2): Here the authors introduce "DOC pools". This does not add to your discussion (DOC pools is not used in a way that is distinct from cluster or source zone in the following discussion) and is confusing to the reader. In the actual riparian zone, these two DOC clusters make up the same DOC pool. Please simplify language and omit the use of pool.

L618 (Conclusions): Example of where "two distinct DOC pools" should be "clusters". The authors assigned wells to be distinct sources/pools, but this delineation of different parts of the DOC pool is defined in their cluster/statistical analyses.

---

## Author Response (AR2)

**Answer to Referee 1:**

Title: Small-scale topography explains patterns and dynamics of dissolved organic carbon exports from the riparian zone of a temperate, forested catchment, by Werner et al.

Werner et al. sent a revised version of a previously submitted manuscript and associated replies to the comments of two previous reviewers. They combined a large number of techniques, namely terrain analyses, field monitoring, sophisticated laboratory work, and a numerical model, to characterize DOC concentrations and chemistry, quantify DOC exports and identify DOC source areas in a (ca. 50 x 50 m) riparian plot of a small temperate forest catchment (although no trees are present in the riparian zone itself). Their main ambition was to shed light on the mechanisms of DOC mobilization and transport in hydromorphic, low relief riparian zones. They conclude that "export by surface runoff" is a primary mechanism and suggest that the TWI can help delineating "the most active source zones".

I was not involved in the previous review round, but I carefully read the revised manuscript and the detailed response letter to the former reviewer comments and found the discussion very interesting and thought-provoking, and I generally support the publication of this article in HESS. The former reviewers already made a detailed assessment of the manuscript and touched on the most important issues from my point of view (I particularly aligned with the comments by reviewer #2). In general, I found that the authors made a good effort addressing the former comments and I think most issues are resolved; thus, I will not provide a very detailed review myself. However, I do have a few further comments that I would like the authors to consider and a particular major point of criticism that was previously raised and that, to me, it requires further clarifications and, potentially, further adjustments in the text and in the interpretations made.

General comments

**R1GC**:
I have a major point of criticism, which was similarly raised by a previous reviewer and which I do not think has been properly addressed (or, at least, I do not fully agree with the authors' view or simply do not completely understand). I am referring to the use of the terms "surface runoff/export" and "overland flow", which the authors claim are the main mechanisms of runoff generation and water delivery to the stream from their riparian zone. The way I (and probably other readers) understand these terms is as water flowing over the soil surface through a continuous path that follows the hydraulic gradient determined by the local slope and eventually discharges into the stream. I guess this definition more or less matches the definition given in L. 439-440 by the authors stating that "surficial runoff is groundwater discharging to the surface or direct precipitation onto saturated areas feeding the stream".

However, in the response letter and in L. 555-556 the authors specify that "with surface runoff we refer to water that has been on the surface at least once on its way to the stream", which to me does not necessarily imply that overland flow is the main mechanism of runoff generation and water delivery to the stream, neither that there is a direct hydrological connection between DOC sources and the stream via overland flow, as it is described in several parts of the manuscript (e.g. L. 564-565, L. 586-588). Indeed, from my interpretation of Figure 5 and Figure S8, the wetter areas with high TWI where surface runoff might be generated are relatively far from the stream and it seems that, even during wet conditions, water would re-infiltrate into the soil in its way to the stream before entering the aquatic system, and therefore the path of direct hydrological connection between the terrestrial and the aquatic compartments would be sub-surficial. This is also something the authors describe as such in the response letter ("Surface flow can reinfiltrate and flow to the stream/ boundaries through the subsurface"). This mechanism makes sense looking at the microtopography of the site and, to me, it is not well described by the authors choice as "surface runoff/export" or "overland flow".

In summary, I am just not comfortable with the use and the highlighted importance that the authors give to surface runoff/overland flow as the main mechanism of runoff generation into the stream from

the riparian zone and I would suggest a reformulation of both the terminology and the description of the mechanism, or a further careful clarification of this issue.

Dear Referee #1,

We appreciate your evaluation of our revised manuscript. We understand your discomfort about our highlighted main mechanism (surface runoff generation), since the argument as stated now does not lead inevitably to our conclusion. Therefore we realized the need of some clarifications that are incorporated in the revised manuscript.
Your comment brings up two issues: 1) our definition of 'surface runoff' in the revision as any flow that had at least one part of its trajectory to the stream above the soil surface; and 2) doubt about overland flow being the main mode of DOC transport to the stream.

We revised our definition of 'surface runoff' since the earlier formulation had apparently generated some confusion instead of clarity (see R1GC1).

The second issue you raise requires a more elaborate response. Field observations confirmed widespread occurrence of overland flow along distinct overland flow networks defined by interconnected depressions of the micro-topography during snow melt and during wet periods in autumn and spring (in winter, the area was covered by snow). The water flowing in those surface flow networks originates from groundwater exfiltrating into the surface depressions as well as direct precipitation onto zones with fully saturated soils (also mainly in the depressions). Although reinfiltration of some of this water along some flow paths would generally be possible where vertical subsurface hydraulic gradients point downward, vertical gradients below the surface flow networks are predominantly upwards during strong rainfall events, when groundwater recharge from the hillslopes and direct precipitation lead to rising groundwater heads in the RZ, in particular below the surface depressions. For the purpose of our study it is neither practical nor necessary to quantify which individual sections of different flow trajectories were above and below the surface, but generally speaking, the flow towards the stream during events was predominantly over the surface (i.e., surface flow in the classical sense, as you define it), but not all trajectories may have had surface flow over their full extent. The volumetric water balance in our model runs confirmed the importance of surface flow as the dominant mechanism to feed the stream. Surface inflows into the channel constituted 66 % of the total flow gain along the simulated stream segment over the simulation period.

Further confirmation that overland flow is the main delivery pathway for DOC to the stream can be found in the chemical fingerprints of individual source areas we sampled. Seven chemical characteristics of these sources (some of which are far from the stream) can be seen in the stream water as well, indicating that the hydrochemistry was not strongly affected when the water was carried from the source to the stream. This is consistent with a rapid delivery via overland flow, but not with a slower flow rate through the chemically active subsurface.
We included this discussion in the text and thank you for helping us sharpen the interpretation of our data (R1GC2).

Specific comments

Abstract

R1C1. Please, add "(DOCII)" after "lower concentrations".
We agree, "$DOC_{II}$" was added in the abstract.

1 Introduction

R1C2. What do you mean when you say that soil water content do not limit DOC mineralization and

production? In the next sentence, you correctly point out that anaerobic conditions lead to low mineralization rates and oxic conditions to high mineralization rates. Even if "large uphill contributing areas deliver a continuous supply of water", soil water content in the RZ will not be constant and mineralization rates will also depend on it.

We agree, we changed the sentence to

Large uphill contributing areas deliver a continuous supply of water to the RZ, leading to generally moist conditions with high groundwater levels, even during dry periods.

R1C3. I don't understand "micro-topography focuses drainage". Please, rephrase this sentence.

We agree, the sentence was changed to
Depressions in micro-topography collect surficial water (Frei et al. 2010, Scheliga et al., 2019). If these puddles grow to connect with each other, continuous but possibly short-lived surface flow channels can develop that can connect hot spots of DOC production in the shallow soil layers of the RZ to the stream and carry DOC to the stream (during so called hot moments).

R1C4. Maybe write "can induce" instead of "induces", as this is one of the factors you are testing in the study.

This was changed in the revised manuscript.

R1C5. I don't think the dominant source layer concept assumes DOC pools to be uniform across the riparian zone. In the cited Ledesma et al. (2015) study, DOC pools and exports were estimated for several riparian profiles and only assumed to be uniform with respect to the grid cell in the DEM where each riparian profile was located, and based on the contributing area of each specific grid cell location.

We agree, the sentence was adapted to

For instance the dominant source layer concept (Ledesma et al., 2015) focuses on depth-dependent differences in DOC pools in distinct soil layers of a boreal catchment.
R1C6. Suggest starting the sentence as "The dominant source layer concept is based on the transmissivity feedback mechanism (Bishop et al., 2004), which accounts…".

We agree, the sentence was changed to

The dominant source layer concept is based on the transmissivity feedback mechanism (Bishop et al., 2004), which accounts for depth-dependent differences in hydraulic conductivities of soils and the resulting changes in the transmissivity of the soil profile under changing groundwater levels.

R1C7. This is the first time an ecoregion (i.e. temperate) is introduced, and I wonder whether some other explicit mentions to temperate or boreal catchments should be included before this point in order to have a more coherent narrative in this sense.

We agree, some other explicit references to temperate or boreal catchments were included before this point in order to have a more coherent narrative. This was implemented in the preceding paragraph:

Currently existing proxies are mainly based on landscape-scale characteristics like different land use types (Pisani et al., 2020), hydromapping based on convergence of topography (Laudon et al., 2016; Ploum et al. 2020) or general topographic wetness (Musolff et al., 2018; Fellman et al., 2017; Andersson and Nyberg, 2009) in boreal and temperate catchments e.g. represented by the topographic wetness index TWI (Beven and Kirkby, 1979)). However, these proxies are still relatively coarse and typically lump the entire RZ into larger spatial units (e.g. model cells). Accordingly, small-scale

heterogeneity of topography and hydrological properties, which can significantly affect the hydrologic connectivity of local source zones to the stream (Frei et al. 2010) are not adequately represented. We argue that refined proxies that explicitly capture the smaller-scale heterogeneity of riparian zones could generally improve our mechanistic understanding of DOC exports from temperate catchments and potentially provide a means to infuse this understanding into DOC export models for larger scales.
…

2 Materials and Methods

R1C8. Could you please report the catchment areas at the inlet and outlet of your study site?

This was added in the MS.

R1C9. Can you report simple stats of the stage-discharge relationship (i.e. R2 and N at least)?

This was added in the MS.

R1C10. These couple of sentences fit better before the sentence starting with "In addition…" in L. 156.

This was changed accordingly.

R1C11. You need to define "FT-ICR-MS" as "Fourier transform ion cyclotron resonance mass spectrometry" at first use.

We agree, FT-ICR-MS was defined.

R1C12. What do you mean by "proper comparability"? April and December samples were the focus of what specific analyses?

We realized that this sentence is unclear. Therefore we changed it to

To ensure a good comparability between sampling dates, we decided to focus on April and December samples, when a complete set of groundwater and surface water data was available.

R1C13. Please, define DI-ESI-MS.

We agree, DI-ESI-MS was defined.

R1C14. Please, define SPE-DOM.

We agree, SPE was defined at first mention.

R1C15. Why was this period selected? I assume a period where substantial variation in groundwater tables is observed would be preferable so to calibrate the model to a wider range of conditions. Was this the case for this period relative to others?

The selected period is a compromise between CPU time, data availability and high variation in groundwater and stream water tables. For clarification, we adjusted the sentence and added an explanation as follows:

A 21-day period (15 November 2017 to 6 December 2017) was selected for the model calibration in view of the high CPU time demand for transient model runs, data availability constraints and data variability requirements. Wet and intermediate conditions generate almost all of the runoff of the riparian zone. The selected calibration period thus incorporates fluctuations during intermediate and

high groundwater situations and therefore covers both system states (from subsurface- to surface-flux dominated).

R1C16. Do you mean "Fig. S6a" and "Fig. S6b" instead of "S3a" and "S3b"?

We agree and apologize for the error. References are to Fig. S3a and b. This was changed in the MS.

R1C17. The two TWI-generated zones are based on the two DOC clusters, right? How?

We realized that there is a sentence missing to explain how TWI zones actually were generated. This was adjusted in the MS:

We applied the Wilcoxon rank sum to test for differences in TWIHR distributions and medians of the two DOC clusters. The median TWI value of the DOCI cluster was used as a manually chosen threshold to separate the RZ into two explicit zones of high and low TWI values. The water balance for the entire model site and the two TWI-generated zones was then estimated and compared to each other between 12 April 2017 and 19 December 2018 by modeling with HydroGeoSphere.

3 Results

R1C18. Was 8.6 °C the mean temperature during 2018 or during the period April 2017-December 2018, as specified in Table 1? If the latter, then it would not make sense to compare this value to an annual mean since the observation period would include two summers and only one winter.

The value "8.6 °C" refers to both the annual mean of the year 2018 and the actual study period. However, in this regard, we realized that Rain and $ET_0$ value aggregations were still on 15 min basis. We apologize for the inconvenience and adjusted the values of Table 1 to the actual study period:

|  | mean | sd | min | max |
|---|---|---|---|---|
| Air temperature [°C] | 8.6 | 8.18 | -18.6 | 33.9 |
| Rain [mm h$^{-1}$] | 0.11 | 0.62 | 0 | 31.8 |
| $ET_0$ [mm d$^{-1}$] | 1.65 | 1.24 | 0 | 4.6 |
| … | … | … | … | … |

R1C19. Wasn't the starting date of the monitoring campaign the 28th February 2017?

Indeed, the starting date of the monitoring campaign was the 28$^{th}$ February 2017. However, it took a while until all groundwater loggers were online. Therefore we decided to start the data analysis and modeling at 12$^{th}$ April 2017.

We clarified this circumstance in section "2.2 The monitoring program":

We carried out intensive field observations from 28 February 2017 until 19 December 2018, and continued the data collection at a lower frequency until 23 July 2019. Note that within this campaign different probes cover partly different measurement periods due to a sequential deployment of the devices. Therefore we decided to set the period for the actual monitoring campaign for data analysis and modeling at 12 April 2017 to 19 December 2018, where the multiparametric dataset is most complete.

4 Discussion
L. 519. Please, remove the repeated "increased flow" words.

The repeated "increased flow" words were removed.

R1C20. There seems to be a small internal contradiction in here. The fact that the association 'high TWI - DOCI class' is related to higher groundwater levels and the association 'low TWI – DOCII class' is related to lower groundwater levels seems to contradict the fact stated in L. 512-513 that "DOC classes were independent of depth". Does this have to do with the distribution of the depths of the piezometer screens, which make it difficult to say anything meaningful about the relationship between DOC characteristics and depth? I have seen some of the explanations you gave around this issue in the response letter and in L. 570-573, but it is still not entirely clear to me how this contradiction is resolved, and I would like if you could include a couple of sentences already in this part of the discussion that clarify this point.

We tried to clarify the text further, although we see no contradiction in the two statements. Sampling/Slot 'depth' is time-independent (see chapter 2.2.3 and Fig. S1 for actual values), whereas groundwater levels are time-dependent. A given depth at a given location can experience low and high groundwater levels at different times:

A differentiation has to be made between mean groundwater levels during the entire study period that we stated to be higher for high TWI zones and the actual groundwater levels at which the sampling happened. The latter varied substantially and allows to take e.g. surface water samples in the low TWI zones (that have lower mean groundwater levels) during high flow situations when the entire RZ is fully saturated. On the other hand, the partly screened piezometers allowed to sample different depths even if the groundwater level was well above the screen. Therefore we argue that the two statements are independent from each other.

This proposition was discussed in the revised manuscript with emphasis on the differences between mean and actual groundwater level.

**Answer to Referee 2:**

Overall, this study is an impressive case study of how, when, and where DOC is exported from the riparian zone in a small headwater catchment.

The number of different field, lab, and modeling techniques employed make this manuscript difficult to follow at times. While much of this difficulty is unavoidable due to the complex nature of the research question, I have made suggestions for the authors to simplify language, particularly around descriptors of their DOC clusters, to help make the intent of their use more clear and purposeful.

As per previous reviewers suggestions, the authors have reworked the introduction and discussion to 1) identify a clear research question or hypothesis and 2) develop a discussion that put the results into perspective. The extensive effort on the author's part to address these comments is commendable and has resulted in a compelling discussion of their results and a well formed hypothesis and introduction. I agree with previous reviewer suggestions that the rationale behind how this study is relevant to management or the argument that DOC export needs to be managed is unclear. I suggest that the authors reframe the first few paragraphs of the introduction to be centered around larger knowledge gaps around linkages between terrestrial-aquatic carbon cycling, transport, and fate.
* * *
Dear Referee,
We appreciate your evaluation of our Manuscript (MS). We realized that we need to keep the language as concise and purposeful as possible in order to allow readers to focus on the textual complexity that is inherent to answering the research question. We addressed this issue specifically by simplifying language around the descriptors of our DOC clusters ("DOC-pool, -type, -source zone, -cluster") in the entire MS. The Referees specific comments constituted a good starting point for doing so (Comment on R2C3, 6, 7, 8, 9). We also reworked the first paragraph of the introduction with regard to the central ecological importance of DOC and its link to management.

This study uses high-resolution field sampling and surface-subsurface hydrologic modelling techniques to determine the spatial and temporal variability in DOC sources and export from a riparian zone. The authors found that two distinct clusters of DOC concentration and composition could be explained by topographic wetness index, which was then used to delineate DOC source zones within the riparian zone. DOC export from high TWI zones was 1.5 times greater than low TWI zones. Overall, this study is an impressive case study of how, when, and where DOC is exported from the riparian zone in a small headwater catchment.

The number of different field, lab, and modeling techniques employed make this manuscript difficult to follow at times. While much of this difficulty is unavoidable due to the complex nature of the research question, I have made suggestions for the authors to simplify language, particularly around descriptors of their DOC clusters, to help make the intent of their use more clear and purposeful.

**R2GC**:
As per previous reviewers suggestions, the authors have reworked the introduction and discussion to 1) identify a clear research question or hypothesis and 2) develop a discussion that put the results into perspective. The extensive effort on the author's part to address these comments is commendable and has resulted in a compelling discussion of their results and a well formed hypothesis and introduction. I agree with previous reviewer suggestions that the rationale behind how this study is relevant to management or the argument that DOC export needs to be managed is unclear. I suggest that the authors reframe the first few paragraphs of the introduction to be centered around larger knowledge gaps around linkages between terrestrial-aquatic carbon cycling, transport, and fate. For these reasons listed above, I suggest that this manuscript be accepted for publication pending minor revisions. I have included line-by-line comments below for specific areas throughout the manuscript.

We thank the reviewer for the positive evaluation of our revisions. Concerning the focus on management we agree with the reviewer that the main contribution of our work is in improving the

understanding of linkages between terrestrial-aquatic carbon cycling, transport, and fate and that management implications can follow from that. We revised the introduction accordingly and scaled back the statements on management implications.

R2C1 (Abstract): This hypothesis does not match the hypothesis in your introduction, or the hypothesis that is referenced throughout the MS.

We agree, the hypothesis in the abstract was adapted to match the hypothesis in the introduction:

Here we show that DOC export is predominantly controlled by the micro-topography of the RZ (lateral variability), and by riparian groundwater level dynamics (temporal variability).

R2C2 (Abstract): Should (n = 66) be (DOCII)?

The sample number refers to the overall riparian samples. We realized that this sentence is ambiguous and therefore changed it to

The chemical classification of the riparian groundwater and surface water samples (n = 66) by Fourier-transform ion cyclotron resonance mass spectrometry revealed a cluster of plant-derived, aromatic, and oxygen-rich DOC with high concentrations (DOCI) and a cluster of microbially processed, saturated, and hetero-atom enriched DOC with lower concentrations (DOCII).

R2C3 (Abstract): Here and elsewhere in the abstract (and main text), "pool", "type", "source zone" and "cluster" are all used in reference to DOCI and DOCII. These descriptors all appear to be used interchangeably, but you are 1) using a cluster analysis to isolate and contrast end members within the broader DOM pool and 2) you are using zones to refer to both DOC and TWI. Also, shouldn't "DOCI source zone with high TWIHR values" be "high TWIHR zones associated with the DOCI cluster", because the zones you are referencing were categorized by TWIHR and then assigned a DOC cluster based on the TWIHR value? I recommend that the authors simplify these descriptors throughout the abstract and MS to just DOC "clusters" to avoid confusion and be representative of the DOC comparison analyses conducted.

We agree with the Reviewer's recommendation and reworked the entire manuscript (MS) in order simplify the DOC terminology. In line with this, the Abstract was changed to
[…]
The chemical classification of the riparian groundwater and surface water samples (n = 66) by Fourier-transform ion cyclotron resonance mass spectrometry revealed a cluster of plant-derived, aromatic, and oxygen-rich DOC with high concentrations (DOCI) and a cluster of microbially processed, saturated, and hetero-atom enriched DOC with lower concentrations (DOCII). The two DOC clusters were connected to locations with distinctly different values of the high-resolution topographic wetness index (TWIHR; @ 1 m resolution) within the study area. Numerical water flow modelling using the integrated surface subsurface model HydroGeoSphere revealed that surface runoff from high TWIHR zones associated with the DOCI cluster (DOCI source zones) dominated overall discharge generation and therefore DOC export. Although corresponding to only 15 % of the area in the studied RZ, the DOCI source zones contributed 1.5 times the DOC export of the remaining 85 % of the area associated with DOCII source zones. Accordingly, DOC quality in stream water sampled under five event flow conditions (n = 73) was closely reflecting the DOCI quality.
[…]

R2GC (Introduction): This first introduction paragraph/section needs more detail and evidence to build the argument that DOC is important. DOC in streams and rivers is of central ecological importance to what? The argument in this paragraph does not support the claim that DOC export needs to be managed and this study does not address questions in which a "for management" framing seems appropriate. More generally, what I think this study does do is use an impressive high resolution field and modeling approach to ask how, when, and where is DOC entering the stream from the riparian

zone. DOC generation, understanding how DOM changes and moves within and across ecosystem interfaces, and linking aquatic and terrestrial carbon cycling are still large knowledge gaps that are 1) needed to then argue for DOC export management and 2) knowledge gaps that this study is addressing! I would suggest returning to the Cole et al. 2007 paper you cite to help reframe this first section of the introduction. I've also included a few citations below of recent papers to help frame this argument:

-Butman D., R. Striegl, S. Stackpoole, P. del Giorgio, Y. Prairie, D. Pilcher, P. Raymond, F. Paz Pellat, and J. Alcocer (2018), Chapter 14: Inland waters. In Second State of the Carbon Cycle Report (SOCCR2): A Sustained Assessment Report. U.S. Global Change Research Program, Washington, DC, USA. 568-595.

-Drake, T. W., P. A. Raymond, and R. G. M. Spencer (2018) Terrestrial carbon inputs to inland waters: A current synthesis of estimates and uncertainty. Limnology & Oceanography Letters, 3, 132-142.

-Vachon, D., R. A. Sponseller, and J. Karlsson (2021), Integrating carbon emission, accumulation and transport in inland waters to understand their role in the global carbon cycle. Global Change Biology, 27, 719-727. https://doi.org/10.1111/gcb.15448

We agree with the reviewer that the main contribution of our work is in improving the understanding of linkages between terrestrial-aquatic carbon cycling, transport, and fate and that management implications can follow from that. We revised the introduction accordingly and scaled back the statements on management implications. The proposed references were included to build the argument that the fate of DOC across ecosystem interfaces is still a large knowledge gap.

L60-95 (Introduction): After reviewing the author's changes and comments from the last round of review, I wanted to say that this section of the introduction does an excellent job of setting up your study, why it matters, and why its important. Great job!

Thank you, we appreciate this comment.

R2C4 (Methods): Leaving auto-sampled stream water unfiltered and unpreserved for up to 4 days affects both your DOC concentration and the molecular composition. Most short term assessments of biodegradable DOC last 4 days where a significant amount of DOC can be taken up (Catalan et al. 2021 found up to 40% of initial DOC could be consumed with the first 200 hours). Can you address this potential degradation effect in some way? Did you auto-sample the same well or in the stream several days in a row/between trips to collect and filter samples? This degradation effect likely affected each of your samples differently as well, depending on the time left unfiltered as well as the DOM and microbial community composition. Some relevant studies to consider:

-Catalán, N., Pastor, A., Borrego, C.M., Casas-Ruiz, J.P., Hawkes, J.A., Gutiérrez, C., von Schiller, D. and Marcé, R. (2021), The relevance of environment vs. composition on dissolved organic matter degradation in freshwaters. Limnol Oceanogr, 66: 306-320. https://doi.org/10.1002/lno.11606

-D'Andrilli, J., Junker, J.R., Smith, H.J. et al.DOM composition alters ecosystem function during microbial processing of isolated sources. Biogeochemistry142, 281–298 (2019).https://doi.org/10.1007/s10533-018-00534-5

Collecting samples right after storm events was practically not feasible due to the remoteness of the study site. Thus, the potential degradation effect before sample treatment constitutes a limitation of our study that was addressed as follows:

Due to the remoteness of the study site, we collected auto-sampled stream water samples within 4 days after the triggered event sampling. Samples were stored in the dark inside the sampler and air temperature was always below 10°C during that time. We are aware that the delayed sample retrieval constitutes a limitation of our study which may affect DOC concentration and composition, in particular with respect to labile DOC sources, e.g. leaf leachate (Catalán et al. 2021). Yet, Werner et al. 2019 concluded that in-stream processing and biodegradation are likely to be of minor importance at our experimental site. Further, DOC composition typically shifts towards more stable,

allochthonous DOC quality during events (Werner et al., 2019). Hence the major fraction of event-DOC is expected to be unaffected within the first four days (Mostovaya et al., 2016; Catalán et al., 2021).

Additional References:

Mostovaya, A., Koehler, B., Guillemette, F., Brunberg, A.-K., and Tranvik, L. J.: Effects of compositional changes on reactivity continuum and decomposition kinetics of lake dissolved organic matter, Journal of Geophysical Research: Biogeosciences, 121, 1733-1746, https://doi.org/10.1002/2016JG003359, 2016.

Catalán, N., Pastor, A., Borrego, C.M., Casas-Ruiz, J.P., Hawkes, J.A., Gutiérrez, C., von Schiller, D. and Marcé, R. (2021), The relevance of environment vs. composition on dissolved organic matter degradation in freshwaters. Limnol Oceanogr, 66: 306-320. https://doi.org/10.1002/lno.11606

R2C5 (Results 3.2.1): This is a clear description of the DOCI cluster, but you as need one for the DOCII as well. Also another reminder to be clear and purposeful with the terms used to describe your DOC clusters (this section is clear and the use of clusters is deliberate).

We agree to parts, as we already made it clear that we are directly contrasting the DOC clusters in this sentence and the sentence before. Therefore we explicitly mention the comparison with the $DOC_{II}$ cluster:

In contrast to the DOCII cluster, samples belonging to the DOCI cluster had higher DOC concentration and their molecular composition was characterized by more oxidized (higher NOSC and waOC), more aromatic molecules (higher waAI), with a lower fraction of heteroatoms (smaller waSC, waNC not shown), and a lower molecular weight (smaller wamz).

R2C6 (Results 3.2.2): This is a clear definition of DOCI and DOCII source zones and agree that following this point, these terms can be used. I also appreciate the parenthetic reminders in the results and discussion (i.e., "high TWI zones"). However, because this definition is buried in the results, I suggest reworking your abstract to be clear around source zones vs. clusters.

We agree, we reworked the abstract accordingly.

R2C7 (Results 3.3): Are "DOC source wells" the same as "DOC source zones"? Maybe change to "wells in DOC source zones" to be more clear?

We agree, this was changed as suggested.

Discussion: I wanted to commend the authors on restructuring their discussion! The discussion is distinct from the results, provides context and explanation of key findings, and stresses the importance of the work (all of which were recommendations made by previous reviewers).

Thank you, we appreciate this comment.

R2C8 (Discussion 4.2): Here the authors introduce "DOC pools". This does not add to your discussion (DOC pools is not used in a way that is distinct from cluster or source zone in the following discussion) and is confusing to the reader. In the actual riparian zone, these two DOC clusters make up the same DOC pool. Please simplify language and omit the use of pool.

We agree, we simplified these DOC descriptors throughout the MS (also suggested in Comment on R2C3, R2C9) to just DOC "clusters" where appropriate.

However, we kept "pool" in situations, where it appears more accurate in the context:

When the combination of spatial and chemical analysis indicates two different DOC pools: one that can be depleted and one that does not (here "cluster" or "source zone" are not adequate). This is accordingly marked in the manuscript.

R2C9 (Conclusions): Example of where "two distinct DOC pools" should be "clusters". The authors assigned wells to be distinct sources/pools, but this delineation of different parts of the DOC pool is defined in their cluster/statistical analyses.

We agree, this was changed (also in line with comment R2C8) according to

The chemical classification of riparian water samples via ultra-high resolution FT-ICR-MS revealed two distinct DOC clusters (DOCI and DOCII) in the riparian zone. Degrading plant material presumably contributes most to an aromatic, oxygen-rich DOC with high concentrations (DOCI cluster), located in regions of high wetness in local topographic depressions. The DOCI is available for photo-degradation, […]